# Intestinal Atp8b1 dysfunction causes hepatic choline deficiency and steatohepatitis

Ryutaro Tamura[1,12], Yusuke Sabu[1,12], Tadahaya Mizuno [1], Seiya Mizuno[2], Satoshi Nakano[3], Mitsuyoshi Suzuki [3], Daiki Abukawa[4], Shunsaku Kaji[5], Yoshihiro Azuma[6], Ayano Inui[7], Tatsuya Okamoto[8], Seiichi Shimizu[9], Akinari Fukuda[9], Seisuke Sakamoto[9], Mureo Kasahara [9], Satoru Takahashi [2], Hiroyuki Kusuhara[1], Yoh Zen[10], Tomohiro Ando [11] & Hisamitsu Hayashi [1] ✉

Choline is an essential nutrient, and its deficiency causes steatohepatitis. Dietary phosphatidylcholine (PC) is digested into lysoPC (LPC), glycerophosphocholine, and choline in the intestinal lumen and is the primary source of systemic choline. However, the major PC metabolites absorbed in the intestinal tract remain unidentified. ATP8B1 is a P4-ATPase phospholipid flippase expressed in the apical membrane of the epithelium. Here, we use intestinal epithelial cell (IEC)-specific Atp8b1-knockout (Atp8b1[IEC-KO]) mice. These mice progress to steatohepatitis by 4 weeks. Metabolomic analysis and cell-based assays show that loss of Atp8b1 in IEC causes LPC malabsorption and thereby hepatic choline deficiency. Feeding choline-supplemented diets to lactating mice achieves complete recovery from steatohepatitis in Atp8b1[IEC-KO] mice. Analysis of samples from pediatric patients with *ATP8B1* deficiency suggests its translational potential. This study indicates that Atp8b1 regulates hepatic choline levels through intestinal LPC absorption, encouraging the evaluation of choline supplementation therapy for steatohepatitis caused by ATP8B1 dysfunction.

Choline, an essential human nutrient, is needed to maintain normal cell function and structure[1,2]. Choline deficiency limits hepatic phosphatidylcholine (PC) synthesis through the CDP-choline pathway, which causes a failure to package and export very low-density proteins (VLDL) in the liver and thereby steatohepatitis[3–5]. Dietary PC is digested into lysoPC (LPC), glycerophosphocholine (GPC), and choline in the intestinal lumen, which are taken up by the intestinal epithelial cells (IEC) and are the primary source of choline in the body[1,6,7]. However, the major PC metabolites absorbed in the

intestinal tract and used for choline synthesis in the body have yet to be identified[7,8].

P4-ATPases, a subfamily of P-type ATPases, are phospholipid translocases in eukaryotic membranes[9]. These proteins maintain membrane lipid asymmetry by translocating phospholipids from the exocytoplasmic to the cytoplasmic leaflet[10]. This lipid asymmetry plays crucial roles in numerous cellular processes, including cell and organelle shape determination and dynamics, vesicle budding and transport, regulation of membrane protein function, and sensory function[11].

[1]Laboratory of Molecular Pharmacokinetics, Graduate School of Pharmaceutical Science, The University of Tokyo, Tokyo, Japan. [2]Laboratory Animal Resource Center and Trans-Border Medical Research Center, University of Tsukuba, Ibaraki, Japan. [3]Department of Pediatrics, Juntendo University Graduate School of Medicine, Tokyo, Japan. [4]Department of Gastroenterology and Hepatology, Miyagi Children's Hospital, Miyagi, Japan. [5]Department of Pediatrics, Tsuyama-Chuo Hospital, Okayama, Japan. [6]Department of Pediatrics, Yamaguchi University Graduate School of Medicine, Yamaguchi, Japan. [7]Department of Pediatric Hepatology and Gastroenterology, Saiseikai Yokohama City Eastern Hospital, Kanagawa, Japan. [8]Department of Pediatric Surgery, Kyoto University Hospital, Kyoto, Japan. [9]Organ Transplantation Center, National Center for Child Health and Development, Tokyo, Japan. [10]Institute of Liver Studies, King's College Hospital & King's College London, London, UK. [11]Axcelead Drug Discovery Partners, Inc., Fujisawa, Kanagawa, Japan. [12]These authors contributed equally: Ryutaro Tamura, Yusuke Sabu. ✉e-mail: hayapi@mol.f.u-tokyo.ac.jp

The human genome encodes 14 P4-ATPases, and mutations in specific P4-ATPases cause severe diseases, including cholestasis[12], neurological disorders[13,14], and congenital hemolytic anemia[15].

ATP8B1 is a member of the P4-ATPase family. It is expressed at the apical membrane of epithelial tissues, most prominently in the intestine, pancreas, and stomach and, to a much lesser extent, in the liver[16]. Mutations in the gene encoding ATP8B1 cause progressive familial intrahepatic cholestasis type 1 (PFIC1), an extremely rare inherited autosomal recessive liver disease[12]. Its prevalence remains unknown, but its estimated incidence varies between 1/50,000 and 1/100,000 births[17]. PFIC1 patients present with severe intrahepatic cholestasis with sustained intractable itching, jaundice, and failure to thrive, resulting in liver failure. They exhibit extrahepatic symptoms, including diarrhea, pancreatic insufficiency, pneumonia, and sensorineural deafness[17–19]. Although several pathogenic models focusing on hepatic ATP8B1 function have been proposed for cholestasis in PFIC1[9,20–22], no effective therapy has yet been developed. Liver transplantation (LTx) solves cholestasis and liver failure in PFIC1[18]. However, it often results in insufficient clinical outcomes in PFIC1 because of exacerbation of steatosis and fibrosis in the graft liver and retardation of stature and weight[23,24]. This indicates that extrahepatic ATP8B1 function contributes to normal liver function, the understanding of which is essential for developing an effective therapy for PFIC1.

Herein, we generated IEC-specific Atp8b1 knockout (Atp8b1[IEC-KO]) mice because the physiological function of Atp8b1 in IEC is unknown despite its high expression levels in IEC[16]. The Atp8b1[IEC-KO] mice showed LPC malabsorption, systemic deficiency in choline and its related metabolites, and steatohepatitis. To pursue the causal relationship behind this, we tested a choline-supplemented diets (CSD) and confirmed its efficacy for alleviating steatohepatitis in Atp8b1[IEC-KO] mice. Finally, plasma from PFIC1 patients was analyzed to examine the translational potential of the findings in Atp8b1[IEC-KO] mice.

## Results

### Loss of Atp8b1 in IEC causes high infant mortality and growth retardation

Two lines of Atp8b1 [flox/flox]; villin-Cre (Atp8b1[IEC-KO]) mice (lines #12 and #51) were generated based on polymerase chain reaction (PCR) genotyping (Supplementary Fig. 1a, b). Quantitative PCR (qPCR) analysis confirmed loss and normal expression of Atp8b1 mRNA in IEC and liver, respectively, of Atp8b1[IEC-KO] mice (line #12) (Fig. 1a). The targeted allele showed normal Mendelian segregation (Supplementary Table 1). However, the loss of Atp8b1 in IEC led to a high infant mortality rate (Fig. 1b). Atp8b1[IEC-KO] mice (line #12) begin to die when they are weaned around 3 or 4 weeks of age, and all die within 12 weeks after birth. Atp8b1[IEC-KO] mice (line #12) showed the normal gross appearance of the whole body and small intestine (SI) when newborn (5 days after birth), but developed growth retardation and elongation of the SI by 4 weeks of age (Fig. 1c, d). At this age, Atp8b1[IEC-KO] mice (line #12) had significantly lower body weight and greater length and weight of the SI than their littermate Atp8b1[flox/flox] mice, whereas no difference in these parameters was observed between both sets of mice when newborn (Fig. 1e–g). Liver weight was increased by the loss of Atp8b1 in IEC at 4 weeks of age, but not at newborns (Fig. 1h). These phenotypes, shown in Fig. 1, were also observed in the other line of Atp8b1[IEC-KO] mice (line #51) (Supplementary Fig. 2).

### Loss of Atp8b1 in IEC causes abnormal morphology of the SI

To obtain insight into the effect of Atp8b1 on the structure and function of the SI, we performed a histological comparison of SI sections from Atp8b1[IEC-KO] mice (line #12) with those from the littermate Atp8b1[flox/flox] mice. Hematoxylin and eosin (H&E) staining showed that no significant difference between the mice was observed at the newborn stage. By contrast, at 4 weeks of age, Atp8b1[IEC-KO] mice (line #12) exhibited shorter villi in the distal SI than Atp8b1[flox/flox] mice (Fig. 2a, b).

Immunohistochemical (IHC) analysis revealed that Atp8b1[IEC-KO] mice (line #12) at 4 weeks of age had lower levels of apical proteins (NHE3, Ezrin, DPPIV, and pERM) and normal basolateral protein levels (E-cadherin and β-catenin) in IEC of the proximal and distal SI, compared with littermate Atp8b1[flox/flox] mice (Fig. 2c–f). When newborn, no apparent differences between the mice were observed in the expression of these proteins. Intestinal epithelium comprises several cell types, including Paneth cells, goblet cells, neuroendocrine cells, and enterocytes[25,26]. The villus cell population was not significantly different between the mice (Fig. 2g–i). Similar results, shown in Fig. 2, were obtained with the other line of Atp8b1[IEC-KO] mice (line #51) (Supplementary Fig. 3).

### IEC without Atp8b1 develop steatohepatitis

At 4 weeks of age, blood levels of aspartate aminotransferase (AST), alanine aminotransferase (ALT), total bilirubin (T-bil), and direct bilirubin (D-bil) were significantly higher in Atp8b1[IEC-KO] mice (line #12) than in littermate Atp8b1[flox/flox] mice. In contrast, at the newborn stage, there was no difference in these values between the mice (Fig. 3a–d). This indicates that liver injury develops in Atp8b1[IEC-KO] mice (line #12) by 4 weeks of age. To investigate the cause of this liver injury, the liver histology of Atp8b1[IEC-KO] mice (line #12) and littermate Atp8b1[flox/flox] mice was assessed. H&E staining showed moderate accumulation of large lipid droplets, mainly in the central lobules in the liver of Atp8b1[IEC-KO] mice (line #12) at 4 weeks of age (Fig. 3e). Consistent with this finding, Plin2, a constitutively associated cytoplasmic lipid droplet coat protein[27], was more intensely stained in the liver of Atp8b1[IEC-KO] mice (line #12) than in that of Atp8b1[flox/flox] mice at 4 weeks of age (Fig. 3f, j). At the newborn stage, although aggregates of small mononuclear cells indicative of extramedullary hematopoiesis were observed, no abnormal histological findings were observed in either mouse (Fig. 3e). The stage of liver injury was explored by IHC staining for myeloperoxidase (MPO), F4/80, glial fibrillary acidic protein (GFAP), and α-smooth muscle actin (αSMA), which are markers of neutrophils, total macrophages, early activation of hepatic satellites cells (HSC), and activated HSC, respectively. MPO-, F4/80-, and GFAP-stained areas were much more significant in Atp8b1[IEC-KO] mice (line #12) than in Atp8b1[flox/flox] mice at 4 weeks of age (Fig. 3g–i, k–m), indicating the infiltration of neutrophils and macrophages and enhancement of liver inflammation in Atp8b1[IEC-KO] mice (line #12) over time. No alpha-SMA-stained area was detected in either mouse. These findings indicated that Atp8b1[IEC-KO] mice (line #12) progressed to steatohepatitis, but not to fibrosis, by 4 weeks of age. Similar results, shown in Fig. 3, were obtained with the other line of Atp8b1[IEC-KO] mice (line #51) (Supplementary Fig. 4).

### IEC without Atp8b1 exhibit aberrant choline metabolism

We inferred that Atp8b1[IEC-KO] mice had nutrient malabsorption because of high infant mortality rate (Fig. 1b and Supplementary Fig. 2b), growth retardation (Fig. 1c and Supplementary Fig. 2c), and reduced expression of apical membrane proteins (Fig. 2c–f and Supplementary Fig. 3c–f). To reveal the mechanism underlying the abnormal phenotypes in Atp8b1[IEC-KO] mice, we conducted metabolomic and lipidomic analyses of IEC, plasma, and liver from Atp8b1[IEC-KO] mice (line #12) and the littermate Atp8b1[flox/flox] mice at 4 weeks of age.

Lipidomic data obtained from IEC were subjected to enrichment analysis of lipid species, which scored the magnitude of difference in each lipid content between Atp8b1[IEC-KO] mice (line #12) and the littermate Atp8b1[flox/flox] mice and determined the trends in lipid changes in Atp8b1[IEC-KO] mice (line #12). The results revealed an increase in LPC in IEC of Atp8b1[IEC-KO] mice (line #12) (Fig. 4a). Regardless of the length and degree of unsaturation of fatty acid, almost all LPC species detected by the lipidomic analysis were elevated in IEC of Atp8b1[IEC-KO] mice (line #12) (Fig. 4b). This finding was confirmed by absolute quantification of LPC species in the brush border membrane fractions prepared from IEC (Supplementary Fig. 5a, b). Metabolomic

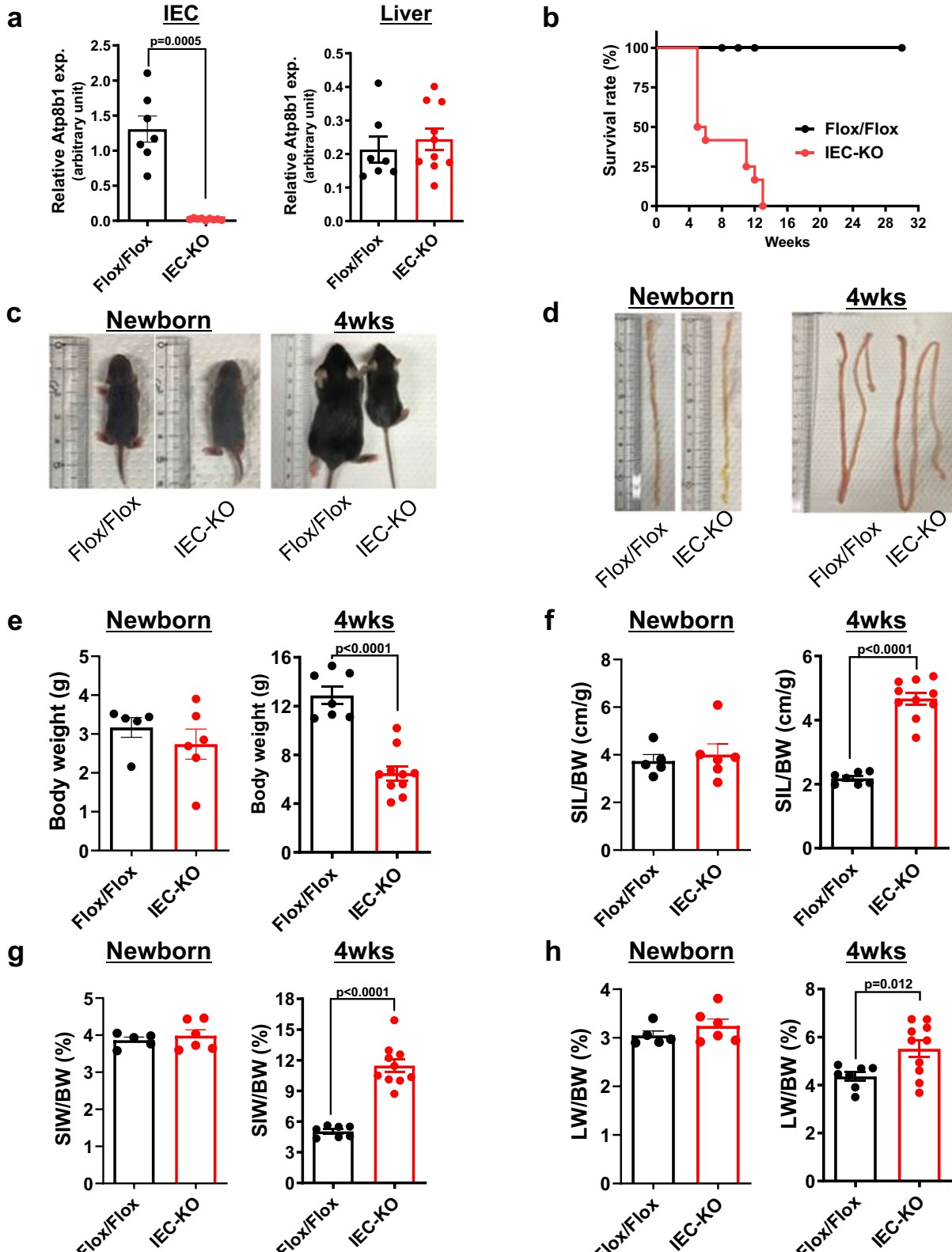

**Fig. 1 | Atp8b1$^{IEC-KO}$ mice (line #12) show a high infant mortality rate, growth retardation, and elongation of the SI. a** Atp8b1 mRNA levels in IEC and liver from male 4-week-old Atp8b1$^{IEC-KO}$ mice (line #12) ($n = 12$) and littermate Atp8b1$^{flox/flox}$ mice ($n = 9$). mRNA levels are expressed relative to those of 18 S rRNA. **b** Survival rate of male Atp8b1$^{IEC-KO}$ mice (line #12) and littermate Atp8b1$^{flox/flox}$ mice ($n = 12$–13 in each group). **c**–**h** Gross appearance of the whole body (**c**) and SI (**d**), BW (**e**), SIL to BW (**f**), SIW to BW (**g**), and LW to BW (**h**) in male newborn and 4-week-old Atp8b1$^{IEC-KO}$ mice (line #12) ($n = 6$ for newborn, $n = 10$ for 4wks) and littermate Atp8b1$^{flox/flox}$ mice ($n = 5$ for newborn, $n = 10$ for 4wks). All data are presented as mean ± SEM. $P$ values were calculated by two-tailed, unpaired Welch's $t$ test and indicated in the figures if less than 0.05. BW body weight, IEC intestinal epithelial cells, LW liver weight, SIW small intestine weight, SIL small intestine length.

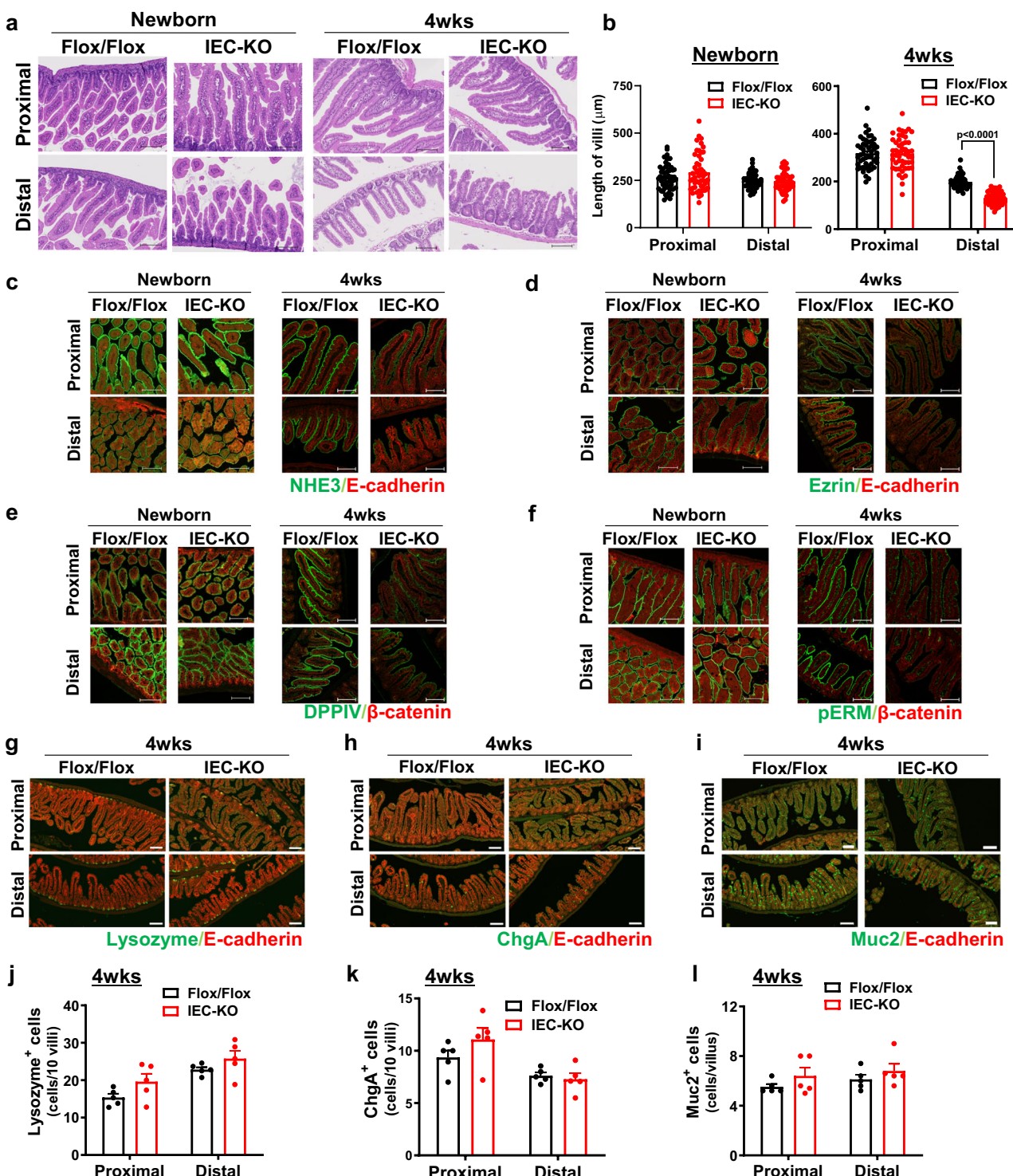

**Fig. 2 | Atp8b1[IEC-KO] mice (line #12) have shortened villi and lower expression of apical membrane protein in SI.** Proximal and distal SI were excised from male Atp8b1[IEC-KO] mice (line #12) and littermate Atp8b1[flox/flox] mice at the newborn stage ($n = 5$) and at 4 weeks old ($n = 5$) and subjected to histological analysis. **a** H&E staining of SI section. **b** Quantification of villus length. Each symbol indicates values of 50 villi from 5 mice in each group. **c–f** IHC staining of apical membrane markers, NHE3 (**c**), ezrin (**d**), DPPIV (**e**), and pERM (**f**), and proteins localized to basolateral membrane, E-cadherin (**c**, **d**) and β-catenin (**e**, **f**), in SI section. **g–l** IHC staining of IEC markers, lysozyme (Paneth cells; **g**), chgA (endocrine cells; **h**), and muc2 (goblet cells; **i**), in SI sections. **j–l** Quantification of IEC positive for lysozyme (**j**), chgA (**k**), and muc2 (**l**) per villus. In each mouse, more than 25 villi were evaluated from three images. Each symbol indicates the values of 5 mice/group. In (**a**, **c–f**, **g–i**), representative images are shown. Scale bars: 100 μm. In (**b**, **j–l**), all data are presented as mean ± SEM. *P* values were calculated by two-tailed, unpaired Welch's *t* test and indicated in the figures if less than 0.05.

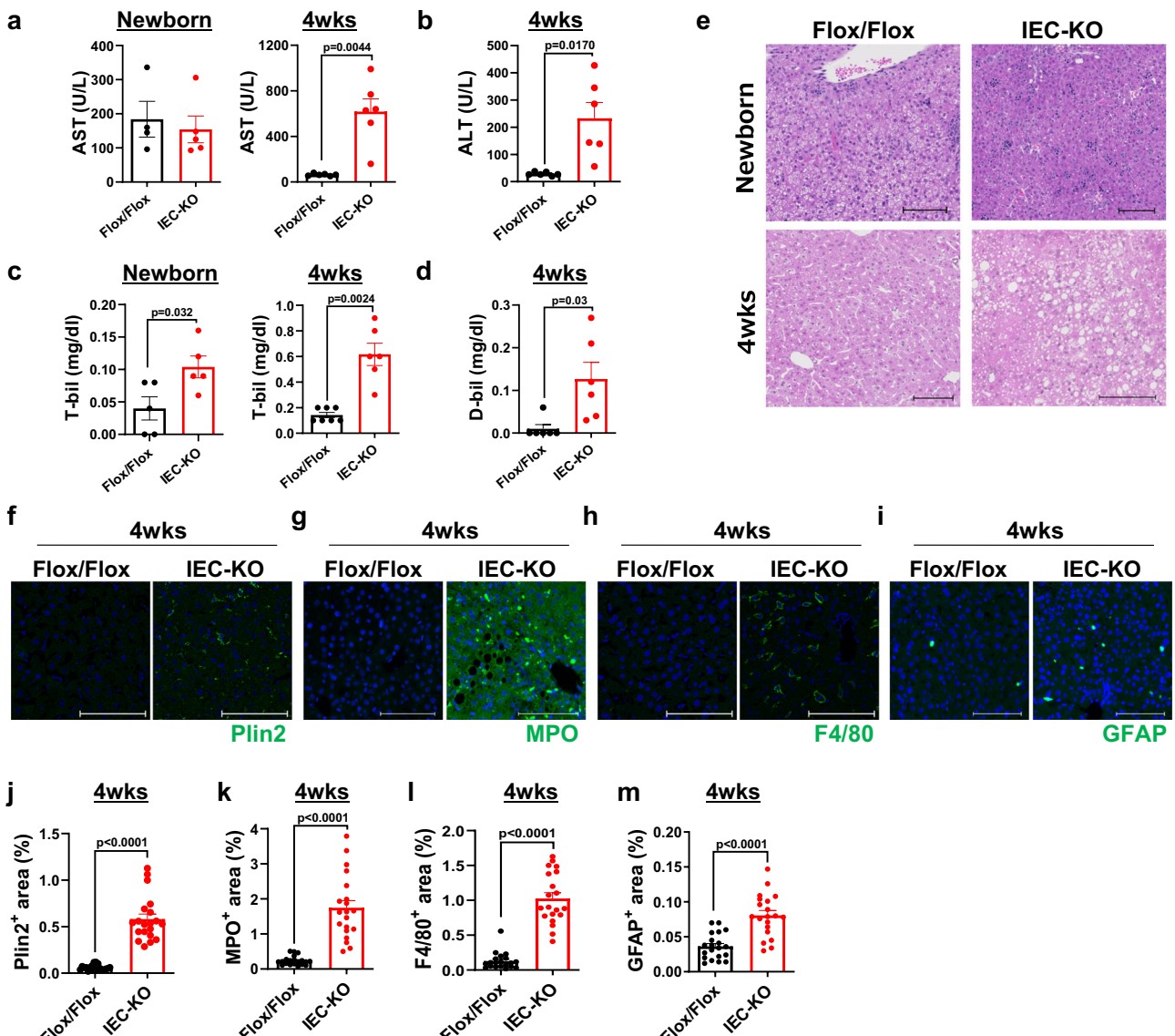

**Fig. 3 | Atp8b1IEC-KO mice (line #12) progress to steatohepatitis.** Plasma and liver were collected from male Atp8b1IEC-KO mice (line #12) and littermate Atp8b1flox/flox mice at the newborn stage (*n* = 5) and at 4 weeks old (*n* = 6) and subjected to biochemical and histological analyses, respectively. **a–d** Plasma biochemistry; AST (**a**), ALT (**b**), T-bil (**c**), and D-bil (**d**). **e** H&E staining of liver section. **f–i** IHC staining of the liver section by antibodies against Plin2 (lipid droplets; **f**), MPO (neutrophils; **g**), F4/80 (macrophages; **h**), and GFAP (quiescent HSC; **i**). **j–m** Quantification of the area stained with Plin2 (**j**), MPO (**k**), F4/80 (**l**), and GFAP (**m**). Each symbol indicates values of 20 images/group from 5 mice/group. In (**e–i**), representative images are shown. Scale bars: 100 μm. In (**a–d**, **j–m**), all data are presented as mean ± SEM. *P* values were calculated by two-tailed, unpaired Welch's *t* test and indicated in the figures if less than 0.05.

analysis showed a trend toward decreased choline metabolites in the plasma and liver of Atp8b1IEC-KO mice (line #12) compared to the littermate Atp8b1flox/flox mice (Fig. 4c–e). The pathway-level analysis of metabolomic data focused on choline metabolism detected a statistically significant deficiency of choline metabolites in the plasma and liver of Atp8b1IEC-KO mice (line #12) (Table 1). Absolute quantification using LC/MS/MS optimized for detecting choline metabolites confirmed that Atp8b1IEC-KO mice (line #12) had lower hepatic levels of choline and its metabolites than the littermate Atp8b1flox/flox mice (Supplementary Fig. 9a).

Enrichment analysis of lipid species (Fig. 4f) using lipidomic data obtained from the liver revealed hepatic TG accumulation in Atp8b1IEC-KO mice (line #12). In addition, biochemical analysis confirmed the increased hepatic TG level in Atp8b1IEC-KO mice (line #12) (Fig. 4g). These findings are consistent with the development of steatohepatitis in these mice (Fig. 3).

## Loss of Atp8b1 in IEC directly affects hepatic choline levels

To confirm whether hepatic choline deficiency was due to a loss of molecular function of Atp8b1, but not the abnormal morphology of the SI resulting from Atp8b1 deficiency, tamoxifen (Tax)-inducible, IEC-specific Atp8b1 knockout (Atp8b1Tax-iIEC-KO) mice (line #12) were generated. qPCR analysis revealed that Tax treatment abolished Atp8b1 mRNA expression in IEC, but not in the liver, of 8-week-old Atp8b1Tax-iIEC-KO mice (line #12) within 6 days (Supplementary Fig. 6a). Atp8b1Tax-iIEC-KO mice (line #12) at 8 weeks of age had no obvious abnormal findings before Tax treatment and up to 6 days after the start of Tax administration. However, on day 14 after the start of Tax administration, they showed significantly lower body weight (Supplementary Fig. 6b), longer SI (Supplementary Fig. 6c), shorter villi in the distal SI (Supplementary Fig. 6e, f), higher AST and ALT values (Supplementary Fig. 7a, b), and more significant accumulation of lipid droplets in the liver (Supplementary Fig. 7c–f) than their littermate

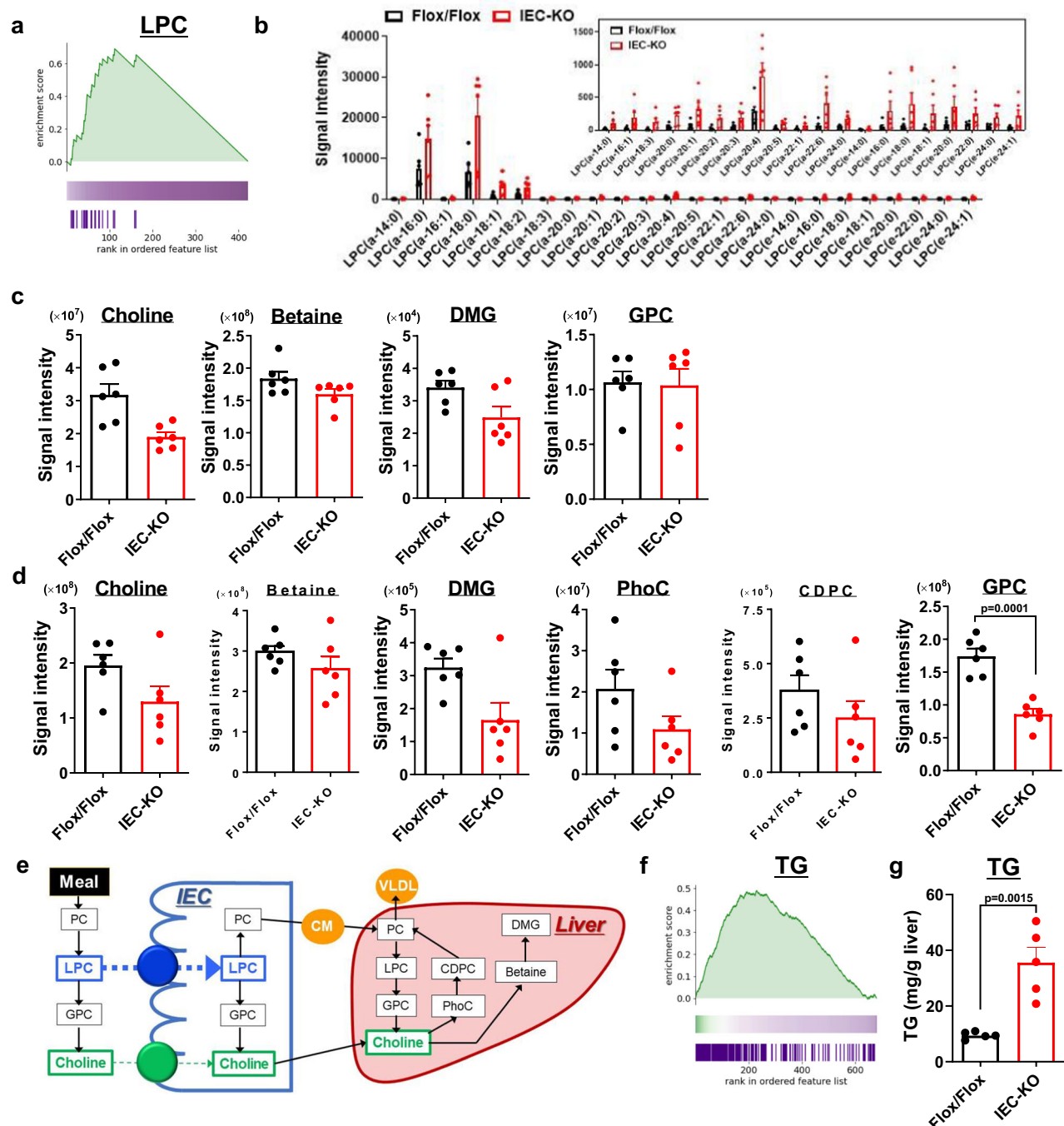

**Fig. 4 | Atp8b1^IEC-KO mice (line #12) show LPC accumulation in IEC and deficiency in choline and its related metabolites in plasma and liver.** IEC, plasma, and liver were collected from male 4-week-old Atp8b1^IEC-KO mice (line #12) and littermate Atp8b1^flox/flox mice (*n* = 6 in each group) and subjected to metabolomic analysis. **a** Enrichment plot for LPC in IEC. The result of the lipidomic analysis was evaluated by Kolmogorov–Smirnov (KS) running sum statistic. The magnitude of difference in each lipid content between Atp8b1^IEC-KO mice and Atp8b1^flox/flox mice was scored as described in Materials and Methods. KS statistics were calculated for LPC and plotted as running sum (top). The middle heatmap and the bottom barcode represent the magnitude of difference in each lipid content and the positions of each LPC species, respectively. **b** The levels of the indicated LPC species in IEC.

**c**, **d** The levels of choline and its metabolites in plasma (**c**) and liver (**d**). **e** Schematic diagram illustrating intestinal digestion and absorption of dietary PC, a primary source of choline in the body. CDPC CDP-choline, CM chylomicron, DMG dimethylglycine, GPC glycerophosphocholine, LPC lysophosphatidylcholine, PC phosphatidylcholine, PhoC phosphorylcholine, VLDL very low-density lipoprotein. **f** Enrichment plot for TG in the liver. The graph was created as described in (**a**). **g** Enzymatic determination of total TG in the liver. In (**b**–**d**, **g**), all data are presented as mean ± SEM. *P* values were calculated by two-tailed, unpaired Welch's *t* test with the Benjamini–Hochberg correction (**d**) or by two-tailed, unpaired Welch's *t* test (**g**) and are indicated in the figures if less than 0.05.

---

Atp8b1^flox/flox mice. These are similar phenotypes with Atp8b1^IEC-KO mice (line #12) (Figs. 1–3). In the liver of Atp8b1^Tax-iIEC-KO mice (line #12), decreased choline levels were observed on day 6 after the start of Tax administration (Supplementary Fig. 7g), suggesting that loss of Atp8b1

in IEC primarily affects hepatic choline levels before intestinal morphology.

These findings suggest that the impaired Atp8b1 function in IEC directly affects hepatic choline levels, limiting PC synthesis through

**Table 1 | Pathway-level analysis to compare the choline metabolites in Atp8b1[IEC-KO] mice (line #12) and the littermate Atp8b1[flox/flox] mice**

| | Integrated $P$ value | Mean difference | Mean $t$ statistics |
|---|---|---|---|
| IEC | 0.11 | 0.56 | 1.47 |
| Plasma | 1.65e−04 | −0.27 | 2.06 |
| Liver | 1.28e−05 | −0.63 | 3.16 |

The metabolome data obtained in Fig. 4 were used for this analysis. Each value was calculated by the generally applicable gene set enrichment (GAGE) method[58]. Briefly, metabolites in the choline pathway and overall metabolites were subjected to Welch's $t$ test, comparing all sample combinations of Atp8b1[IEC-KO] mice (line #12) and the littermate Atp8b1[flox/flox] mice. Negative log-sum values of resultant $P$ values were adjusted based on control group dependencies, and the integrated $P$ value was computed on a Gamma distribution with $K$ degrees of freedom and a scale of 1.0, where $K$ represents the number of samples of Atp8b1[IEC-KO] mice (line #12). *IEC* intestinal epithelial cells.

the CDP-choline pathway, reducing hepatic PC levels, and developing steatohepatitis.

## Atp8b1 flips LPC from exoplasmic to cytosolic leaflets

Considering that (1) ATP8B1 is expressed at the apical membrane of the enterocytes[16], (2) ATP8B1 is a phospholipid flippase[28,29], and (3) Atp8b1[IEC-KO] mice exhibited LPC accumulation in the brush border membrane of IEC (Supplementary Fig. 5), we hypothesized that ATP8B1 mediates LPC absorption through LPC flipping activity at the apical membrane of IEC. Aberrant ATP8B1 function causes LPC accumulation in the exoplasmic leaflet at the apical membrane of IEC and, thereby, LPC malabsorption.

To evaluate this hypothesis, we examined whether ATP8B1 had flippase activity of LPC using CHO-K1 and HEK293T cells with exogenous ATP8B1 expression. The flipping activity of ATP8B1 was assessed by a cell-based flippase assay, in which incorporation into the inner leaflet of the plasma membrane of nitrobenzoxadiazole (NBD)-labeled phospholipids was measured using flow cytometry[30,31]. In both CHO-K1 and HEK293T cells, exogenous ATP8B1 expression increased the levels of bovine serum albumin (BSA) non-extractable NBD-LPC (Fig. 5a and Supplementary Fig. 8a) as well as NBD-PC (Fig. 5b and Supplementary Fig. 8b), a previously reported substrate of ATP8B1[29]. These results indicate that ATP8B1 incorporates NBD-LPC and NBD-PC into the inner leaflet of the plasma membrane. Next, we evaluated the effect of ATP8B1 expression on cytotoxicity to LPC and edelfosine, a synthetic lipase-resistant analog of LPC that exerts cytotoxicity by disrupting lipid rafts[32–34]. Both compounds were added to CHO-K1 and HEK293T cells at different concentrations for 24 h. In both cell lines, exogenous ATP8B1 expression markedly reduced LPC cytotoxicity (Fig. 5c and Supplementary Fig. 8c), suggesting that ATP8B1 promotes LPC flipping into the cytoplasmic leaflet and thereby intracellular metabolism of LPC and confers resistance to LPC. Meanwhile, ATP8B1 had almost no protective effect on edelfosine treatment (Fig. 5d and Supplementary Fig. 8d).

To confirm the dominant contribution of ATP8B1 to LPC flipping in IEC, IEC were prepared from 8-week-old Atp8b1[Tax-ilEC-KO] mice (line #12) and their littermate Atp8b1[flox/flox] mice 6 days after the start of Tax treatment, when Atp8b1[Tax-ilEC-KO] mice (line #12) lose Atp8b1 mRNA expression in IEC and develop choline deficiency but maintain normal appearance of SI. The prepared IEC were analyzed by the flippase assay. The levels of BSA non-extractable NBD-LPC (Fig. 5e), as well as NBD-PC (Fig. 5f), were significantly lower in IEC from Atp8b1[Tax-ilEC-KO] mice (line #12) than those from their littermate. These results indicate that ATP8B1 mediates the incorporation of NBD-LPC and NBD-PC into the inner leaflet of the plasma membrane of IEC.

LPC acyltransferase (Lpcat) could affect LPC levels in IEC because LPC is taken up, reacylated to PC by Lpcat, especially Lpcat3[35,36], and incorporated into chylomicrons in the enterocytes[37]. However, Lpcat

activity in IEC from Atp8b1[IEC-KO] mice (line #12) was comparable to that from littermate Atp8b1[flox/flox] mice (Fig. 5g), indicating no association of LPC accumulation with Lpcat activity in IEC of Atp8b1[IEC-KO] mice.

These findings suggest that LPC flipping by Atp8b1 in IEC plays a dominant role in LPC absorption, thereby maintaining normal levels of systemic and hepatic choline (Fig. 4e).

## Loss of Atp8b1 in IEC does not affect the absorption of choline itself

The primary source of choline is dietary PC[1,6], which is digested to LPC, GPC, or choline in the intestinal lumen[7], taken up by IEC, and used for PC synthesis in the liver through the CDP-choline pathway (Fig. 4e). Atp8b1[IEC-KO] mice (line #12) are considered to have a normal ability to digest PC in the intestinal lumen and absorb choline and GPC in the IEC because daily food intake and fecal content of PC, GPC, and choline were comparable between Atp8b1[IEC-KO] mice (line #12) and the littermate Atp8b1[flox/flox] mice (Supplementary Fig. 9b, c). Normal intestinal absorption and tissue distribution of choline itself in mice with IEC lacking Atp8b1 was confirmed in a study in which [3H]-choline was administered orally; [3H]-radioactivity in SI, liver, and plasma of Atp8b1[Tax-ilEC-KO] mice (line #12) was comparable to the littermate Atp8b1[flox/flox] mice (Supplementary Fig. 9d).

Hepatic phosphatidylethanolamine N-methyltransferase (PEMT) catalyzes methylation of phosphatidylethanolamine (PE) to PC involving methionine and its metabolites and provides the endogenous synthetic pathway of PC. Still, its contribution to the hepatic PC pool is less than 30%[38]. This pathway contributes to endogenous choline production because phospholipase and lysophospholipase can hydrolyze the synthesized PC to produce phosphatidic acid and choline. No significant difference was observed in hepatic Pemt mRNA expression between Atp8b1[IEC-KO] mice (line #12) and the littermate Atp8b1[flox/flox] mice at 4 weeks of age (Supplementary Fig. 10a). LC/MS/MS analysis showed that both mice had similar levels of methionine and its metabolites including S-adenosyl-methionine (SAM), a substrate of the PEMT pathway, in the liver (Supplementary Fig. 10b). The Pemt pathway in Atp8b1[IEC-KO] mice appears to function similarly in littermate Atp8b1[flox/flox] mice.

Considering these results and the nature of the choline metabolic pathway (Fig. 4e), it is feasible that impaired Atp8b1 function in IEC affects LPC absorption, leading to hepatic PC and choline deficiency and the development of steatohepatitis.

## Choline supplementation prevents steatohepatitis in Atp8b1[IEC-KO] mice

Atp8b1[IEC-KO] mice have been suggested to have normal choline absorption in the SI (Supplementary Fig. 9b–d), as an alternative pathway to obtain a systemic choline source (Fig. 4e). Therefore, it is conceivable that choline supplementation therapy prevents the onset of steatohepatitis in Atp8b1[IEC-KO] mice. Because breast milk composition reflects the choline content of the diet[39,40], we tested the feeding of a 1.0% CSD to mothers (line #12) during lactation. The Atp8b1[IEC-KO] pups and Atp8b1[flox/flox] littermates were analyzed immediately after weaning (4 weeks of age). CSD feeding did not improve the survival rate, growth retardation, or greater weight and length of the SI in Atp8b1[IEC-KO] mice (line #12) (Fig. 6a–d), whereas it relieved the heavier liver weight in Atp8b1[IEC-KO] mice (line #12) (Fig. 6e). In a histological analysis of SI sections, CSD feeding had no significant impact on abnormal morphology of SI, including shorter villi of the distal SI in Atp8b1[IEC-KO] mice (line #12) (Fig. 6f, g). Meanwhile, CSD feeding suppressed the elevation of biochemical markers of liver injury, AST, ALT, T-bil, and D-bil, in Atp8b1[IEC-KO] mice (line #12) to the same extent as in Atp8b1[flox/flox] mice (Fig. 6h–k). In Atp8b1[IEC-KO] mice (line #12), decreased levels of hepatic PC due to choline deficiency were corrected by CSD feeding (Fig. 6l). H&E and IHC staining of liver sections showed that CSD feeding resolved the accumulation of lipid droplets (Fig. 6m, n, r),

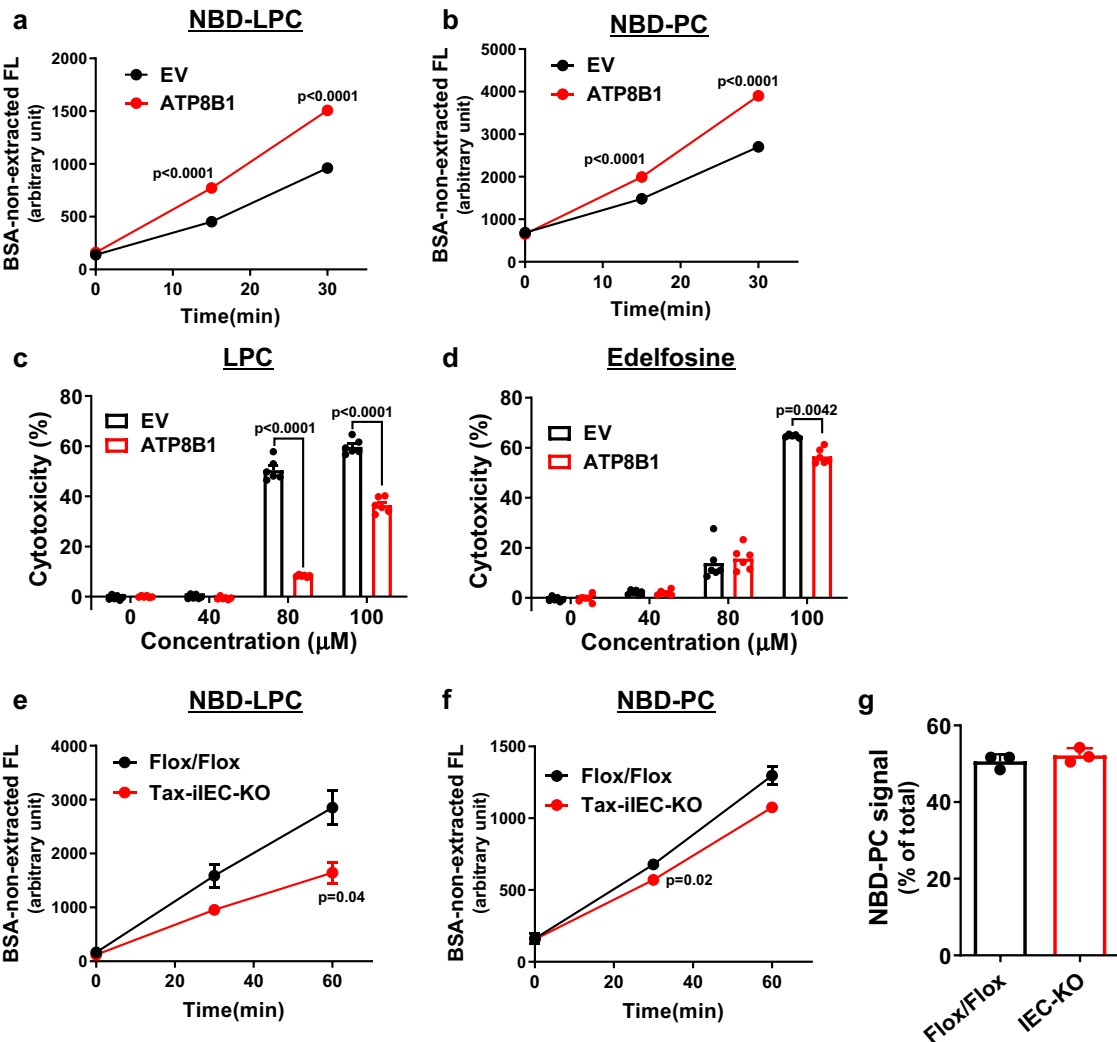

**Fig. 5 | Atp8b1 has flipping activity to LPC. a–d** CHO-K1 cells were transfected with pShuttle-ATP8B1–FLAG or corresponding empty vector and analyzed to evaluate flippase activity to NBD-LPC (**a**) and NBD-PC (**b**) and susceptibility to toxicity of LPC (**c**) and edelfosine (**d**), a synthetic lipase-resistant LPC analog. **e, f** Male 8-week-old Atp8b1[Tax-iIEC-KO] mice (line #12) and littermate Atp8b1[flox/flox] mice were treated daily with 1 mg Tax intraperitoneally for 4 days. IEC were prepared from these mice and analyzed to evaluate flippase activity to NBD-LPC (**e**) and NBD-PC (**f**). As described in Supplementary Materials, for flippase activity measurement (**a, b, e, f**), the cells incubated with NBD-lipids were washed with 5% fatty acid-free BSA to remove NBD-lipids incorporated into the exoplasmic leaflet of the plasma membrane and then analyzed by FACS. Each bar represents the mean ± SEM of quadruple (**a, b**), sextuple (**c, d**), and triplicate (**e, f**) determinations. Where vertical bars are not shown, the SEM is contained within the limits of the symbol. A representative result of two independent experiments is shown. *P* values were calculated by two-tailed, unpaired Welch's t test. EV empty vector. **g** Lpcat activity in IEC. IEC from 4-week-old Atp8b1[IEC-KO] mice (line #12) and littermate Atp8b1[flox/flox] mice (*n* = 3 in each group) were collected, homogenized to prepare Lpcat fraction, and analyzed to evaluate Lpcat activity. Data are represented as mean ± SEM. *P* values were calculated by two-tailed, unpaired Welch's t test and are indicated in the figures if less than 0.05.

infiltration of neutrophils and macrophages (Fig. 6o, p, s, t), and early activation of HSC (Fig. 6q, u) in Atp8b1[IEC-KO] mice (line #12). These results indicated that choline supplementation therapy restores hepatic PC deficiency and completely suppresses the development of steatohepatitis, but does not influence high infant mortality rate, growth retardation, and the abnormal morphology of SI in Atp8b1[IEC-KO] mice.

## Choline and its metabolites are deficient in PFIC1 patients

To examine the translational potential of the findings in Atp8b1[IEC-KO] mice, we conducted a nationwide Japanese survey to identify patients with PFIC1, an extremely rare pediatric cholestatic liver disease resulting from a genetic defect of ATP8B1, and collected their plasma. Twenty-two PFIC1 patients [pre-liver transplantation (LTx), *n* = 10; post-LTx, *n* = 12], 47 patients with other cholestatic diseases (pre-LTx, *n* = 27; post-LTx, *n* = 20), and age-matched control subjects (*n* = 30) were enrolled in this study. The demographic characteristics of the

subjects are outlined in Table 2, Supplementary Tables 2 and 3. Groups of patients with other cholestasis pre-LTx and post-LTx were comprised of normal GGT cholestasis with and without genetic diagnosis (Supplementary Table 2). Groups of patients with PFIC1 and other cholestatic disorders before LTx had similar plasma levels of markers of hepatocellular injury and cholestasis. Consistent with previous reports[23,24], plasma levels of markers of cholestasis, but not hepatocellular injury, were relieved in the PFIC1 patients after LTx. Both values were within normal ranges in patients with other cholestatic disorders after LTx. Hepatic steatosis was confirmed in all PFIC1 post-LTx patients, but not in PFIC1 pre-LTx patients (Supplementary Table 3).

LC/MS/MS analysis showed that plasma levels of choline, betaine, and DMG were significantly lower in PFIC1 patients than in patients with other cholestatic diseases and age-matched control subjects (Fig. 7a–c). Choline deficiency can cause a lack of PC and disrupt VLDL packaging and secretion in the liver, resulting in steatohepatitis[3–5]. Plasma VLDL profiles were characterized using gel filtration high-

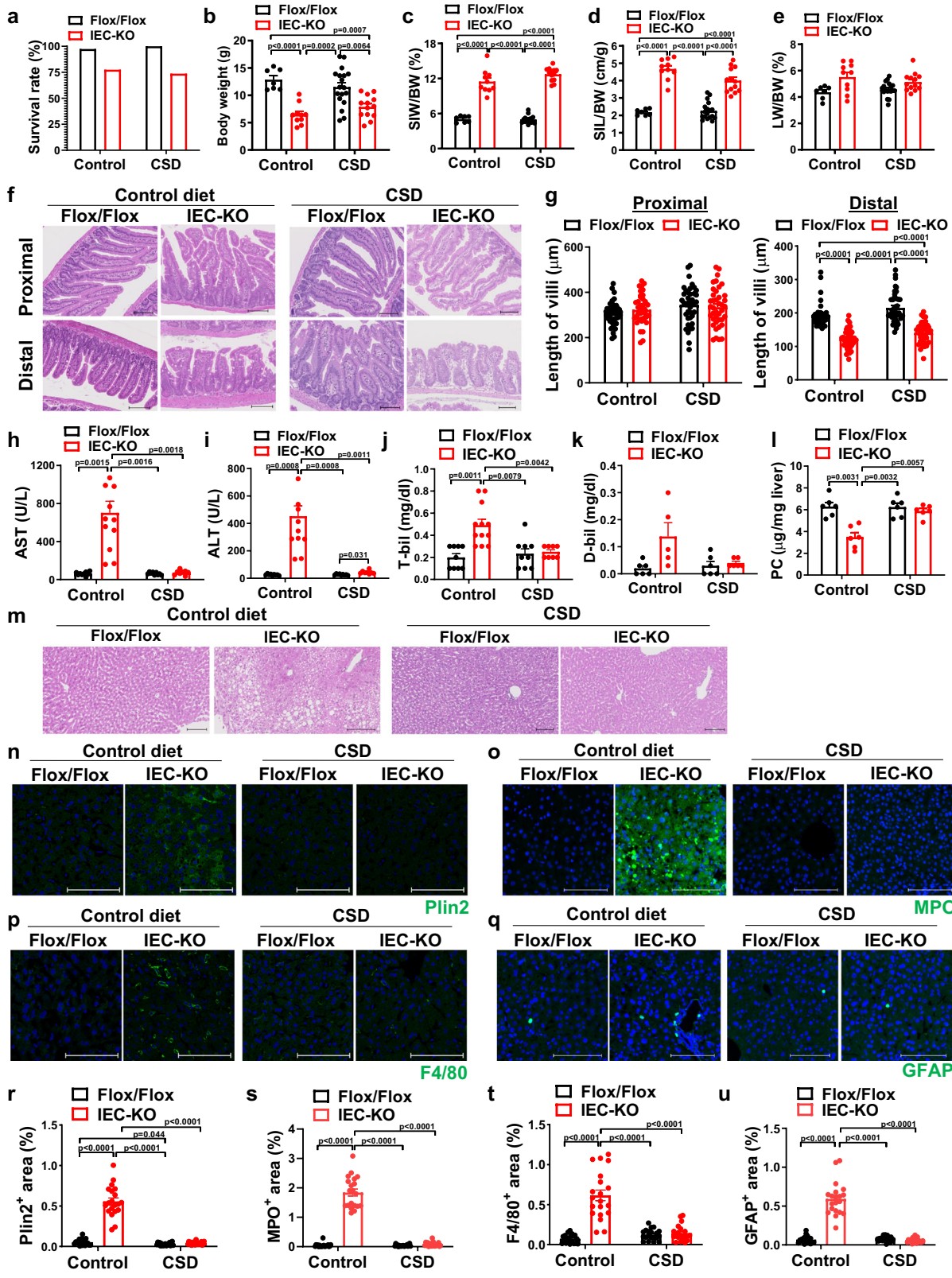

performance liquid chromatography. TG and cholesterol contents in VLDL were decreased in post-LTx PFIC1 patients, but not in pre-LTx PFIC1 patients, compared with the levels in patients with other cholestatic diseases and age-matched control subjects (Fig. 7d, e). VLDL particle size was normal in all groups (Fig. 7f). These results suggest that PFIC1 patients present with deficiency in choline and its metabolites and that they have abnormal hepatic VLDL secretion after LTx,

which is consistent with the fact that steatohepatitis often develops in the graft liver of the PFIC1 patients[23,24].

## Discussion

This study found that Atp8b1[IEC-KO] mice showed LPC malabsorption in IEC, deficiency in choline and its related metabolites in plasma and liver, and subsequent progression to steatohepatitis (Figs. 1–5). As

**Fig. 6 | CSD eliminates steatohepatitis in Atp8b1$^{IEC-KO}$ mice (line #12).** Female mice mated to obtain Atp8b1$^{IEC-KO}$ mice (line #12) were fed a control diet (including 0.1% choline) or 1.0% CSD during lactation. The pups, Atp8b1$^{IEC-KO}$ mice (line #12) and littermate Atp8b1$^{flox/flox}$ mice, were analyzed immediately after weaning (4 weeks old). **a** Survival rate of Atp8b1$^{IEC-KO}$ mice (line #12) and littermate Atp8b1$^{flox/flox}$ mice ($n = 14$–16 in each group). **b–e** BW (**b**), SIW to BW (**c**), SIL to BW (**d**), and LW to BW (**e**) of Atp8b1$^{IEC-KO}$ mice (line #12) ($n = 10$ for control diet, $n = 10$ for CSD) and littermate Atp8b1$^{flox/flox}$ mice ($n = 7$ for control diet, $n = 18$ for CSD). **f** H&E staining of section from proximal and distal SI of Atp8b1$^{IEC-KO}$ mice (line #12) and littermate Atp8b1$^{flox/flox}$ mice. **g** Quantification of villus length. Each symbol indicates values of 40 villi from 5 mice in each group. **h–k** Plasma levels of AST (**h**), ALT (**i**), T-bil (**j**), and D-bil (**k**) of Atp8b1$^{IEC-KO}$ mice (line #12) [$n = 12$ for control diet, $n = 8$ for CSD (**h–j**); $n = 6$ per group (**k**)] and littermate Atp8b1$^{flox/flox}$ mice [$n = 9$ per

group (**h–j**); $n = 6$ per group (**k**)]. **l** PC levels in liver from Atp8b1$^{IEC-KO}$ mice (line #12) and littermate Atp8b1$^{flox/flox}$ mice ($n = 6$ in each group). **m–q** HE (**m**) and IHC (**n–q**) staining of liver section from Atp8b1$^{IEC-KO}$ mice (line #12) and littermate Atp8b1$^{flox/flox}$ mice. Plin2 (lipid droplets; **n**), MPO (neutrophils; **o**), F4/80 (macrophages; **p**), and GFAP (quiescent HSC; **q**) were stained. **r–u** Quantification of the area stained with Plin2 (**r**), MPO (**s**), F4/80 (**t**), and GFAP (**u**). Each symbol indicates values of 20 images/group from 5 mice/group. In (**f**, **m–q**), representative images are shown. Scale bars: 100 μm. In (**b–e**, **g–l**, **r–u**), all data are presented as mean ± SEM. $P$ values were calculated by Welch's one-way ANOVA with a post hoc Dunnett's T3 test for multiple comparisons and are indicated in the figures if less than 0.05. BW body weight, CSD choline supplemental diet, IEC intestinal epithelial cells, LW liver weight, SIW small intestine weight, SIL small intestine length.

**Table 2 | Characteristics and biochemistry of patients at the time of blood sampling**

| | Control subjects ($n = 31$) | PFIC1 pre-LTx ($n = 10$) | PFIC1 post-LTx ($n = 12$) | Other cholestasis pre-LTx ($n = 27$) | Other cholestasis post-LTx ($n = 20$) |
|---|---|---|---|---|---|
| Year of birth | 2016 (2014 to 2017) | 2012 (2006 to 2014) | 2004 (2000 to 2008) | 2012 (2010 to 2013) | 2010 (2005 to 2013) |
| Male (%) | 56.7 | 50.0 | 41.7 | 48.1 | 55.0 |
| Age (years) | 3.5 (2.1 to 5.5) | 5.5 (1.3 to 6.4) | 12.0 (9.4 to 15.0) | 3.0 (1.5 to 8.7) | 8.9 (6.3 to 11.6) |
| Years after LTx | NA | NA | 7.1 (1.3 to 12.2) | NA | 7.2 (5.3 to 9.1) |
| Height $Z$-score | −0.1 (−0.5 to 0.6) | −3.3 (−4.4 to −2.8) | −3.4 (−4.6 to −3.0) | −0.7 (−1.8 to 0.2) | −0.7 (−1.4 to 0.2) |
| Weight $Z$-score | 0.0 (−0.5 to 0.4) | −2.0 (−2.7 to −1.2) | −1.9 (−2.2 to −1.4) | −1.1 (−1.7 to 0.3) | −0.4 (−0.7 to 0.0) |
| AST (IU/L) | 31.0 (26.3 to 37.8) | 78.5 (65.5 to 134.5) | 52.5 (41.8 to 70.5) | 125.5 (66.0 to 329.8) | 34.0 (30.5 to 40.0) |
| ALT (IU/L) | 13.5 (11.0 to 18.5) | 51.5 (43.0 to 91.8) | 45.0 (30.5 to 80.0) | 145.0 (40.5 to 249.0) | 22.0 (14.0 to 25.0) |
| T-Bil (mg/dL) | 0.4 (0.3 to 0.5) | 5.0 (4.2 to 6.3) | 0.7 (0.5 to 0.8) | 3.1 (2.1 to 6.2) | 0.6 (0.6 to 1.0) |
| D-Bil (mg/dL) | 0.1 (0.1 to 0.1) | 4.0 (3.6 to 4.9) | 0.2 (0.1 to 0.2) | 2.3 (1.0 to 4.8) | 0.2 (0.1 to 0.3) |
| GGT (IU/L) | 10.0 (8.0 to 12.8) | 27.5 (25.0 to 37.8) | 24.0 (17.0 to 80.8) | 25.0 (15.3 to 42.0) | 9.0 (6.0 to 54.5) |
| TBA (μmol/L) | 11.9 (6.1 to 19.5) | 222.5 (198.9 to 245.1) | 23.2 (14.4 to 49.3) | 164.8 (79.6 to 257.8) | 42.6 (32.1 to 67.6) |

Data are presented as medians and interquartile ranges.
*LTx* liver transplantation, *NA* not applicable, *TBA* total bile acids.

expected, a CSD completely suppressed steatohepatitis in Atp8b1$^{IEC-KO}$ mice (Fig. 6). These results suggest that choline in the body could be dominantly supplied by the absorption of LPC digested from dietary PC. The translational potential of these findings was suggested by the observation that patients with PFIC1 had lower plasma concentrations of choline and its related metabolites than age-matched control subjects and other cholestatic patients (Fig. 7).

The fatty acyl chains of PC are hydrolyzed into LPC by PLA2 in the lumen, taken up in the enterocytes, and reacylated by Lpcat, in a process called the Lands' cycle[37,41,42]. ATP8B1 is expressed at the apical membrane of IEC[16] and had flippase activity toward LPC (Fig. 5a). Together with the finding of LPC accumulation in the brush border membrane of IEC in Atp8b1$^{IEC-KO}$ mice (Supplementary Fig. 5), LPC malabsorption in Atp8b1$^{IEC-KO}$ mice could be explained by LPC accumulation at the exoplasmic leaflet of the apical membrane of IEC because of the impaired transport of LPC from the exoplasmic leaflet to the inner leaflet. In the intestine, Atp8b1 has a critical role in the Lands' cycle by facilitating the delivery of LPC to Lpcat3, the most highly expressed Lpcat in the intestine that produces arachidonoyl phospholipids[35,36,43], and thereby contributes to PC and choline homeostasis. Lpcat3-knockout mice presented with phenotypes similar to those of Atp8b1$^{IEC-KO}$ mice, such as abnormal intestine morphology and steatohepatitis, implying a functional relationship between Atp8b1 and Lpcat3 in the intestine[35,44].

In humans, mutations in the gene encoding ATP8B1 cause PFIC1, an inherited autosomal recessive liver disease with severe intrahepatic cholestasis[12], which eventually requires LTx. In PFIC1, LTx can improve cholestasis, but patients often develop allograft steatosis, progressing to steatohepatitis and cirrhosis[23,24]. The molecular mechanism of allograft steatosis remains to be elucidated. Herein, we have shown

that the most likely mechanism is deficiency in choline and its related metabolites (Fig. 7a–c) because choline deficiency causes steatosis mainly because of impairment of VLDL secretion associated with insufficient availability of PC[3–5]. We confirmed decreased plasma levels of TG and cholesterol contents in VLDL in post-LTx PFIC1 patients (Fig. 7d, e). These findings also make it possible to explain that, despite the choline deficiency (Fig. 7a–c), pre-LTx PFIC1 patients had normal plasma VLDL (Fig. 7d–f) and did not develop steatosis[17,23,45,46]. Pre-LTx PFIC1 patients suffer from severe cholestasis, a disease state characterized by decreased bile flow because of impaired bile secretion of bile acids, PC, and cholesterol[47], resulting in hepatic accumulation of bile constituents, including PC[48,49]. Additionally, PC metabolism is changed under cholestatic conditions to maintain a normal PC/PE ratio, preventing steatohepatitis[50]. Thus, in pre-LTx PFIC1 patients, severe cholestasis protectively alters PC dynamics against the lack of hepatic PC due to dysfunction of ATP8B1 in the IEC and subsequent choline deficiency and suppresses the development of steatohepatitis. Therefore, choline deficiency occurs in both pre-LTx and post-LTx PFIC1 patients, but only in post-LTx PFIC1 patients is the hepatic PC level below the lower limit able to maintain normal VLDL secretion, leading to the development of steatosis (Supplementary Fig. 11). This is supported by previous findings that Atp8b1$^{G308V/G308V}$ mice homozygous for an ortholog of the ATP8B1 mutant in PFIC1 patients exhibit cholestasis, but not steatosis[28,51]. Atp8b1$^{G308V/G308V}$ mice are genetically deficient in Atp8b1 function throughout the body, including the SI and liver, and are animal models for Pre-LTx PFIC1. On the other hand, Atp8b1$^{IEC-KO}$ and Atp8b1$^{Tax-iIEC-KO}$ mice established in this study are compatible animal models of post-LTx PFIC1.

CSD could not relieve the growth retardation and high infant mortality rate in Atp8b1$^{IEC-KO}$ mice, indicating that deficiency of choline

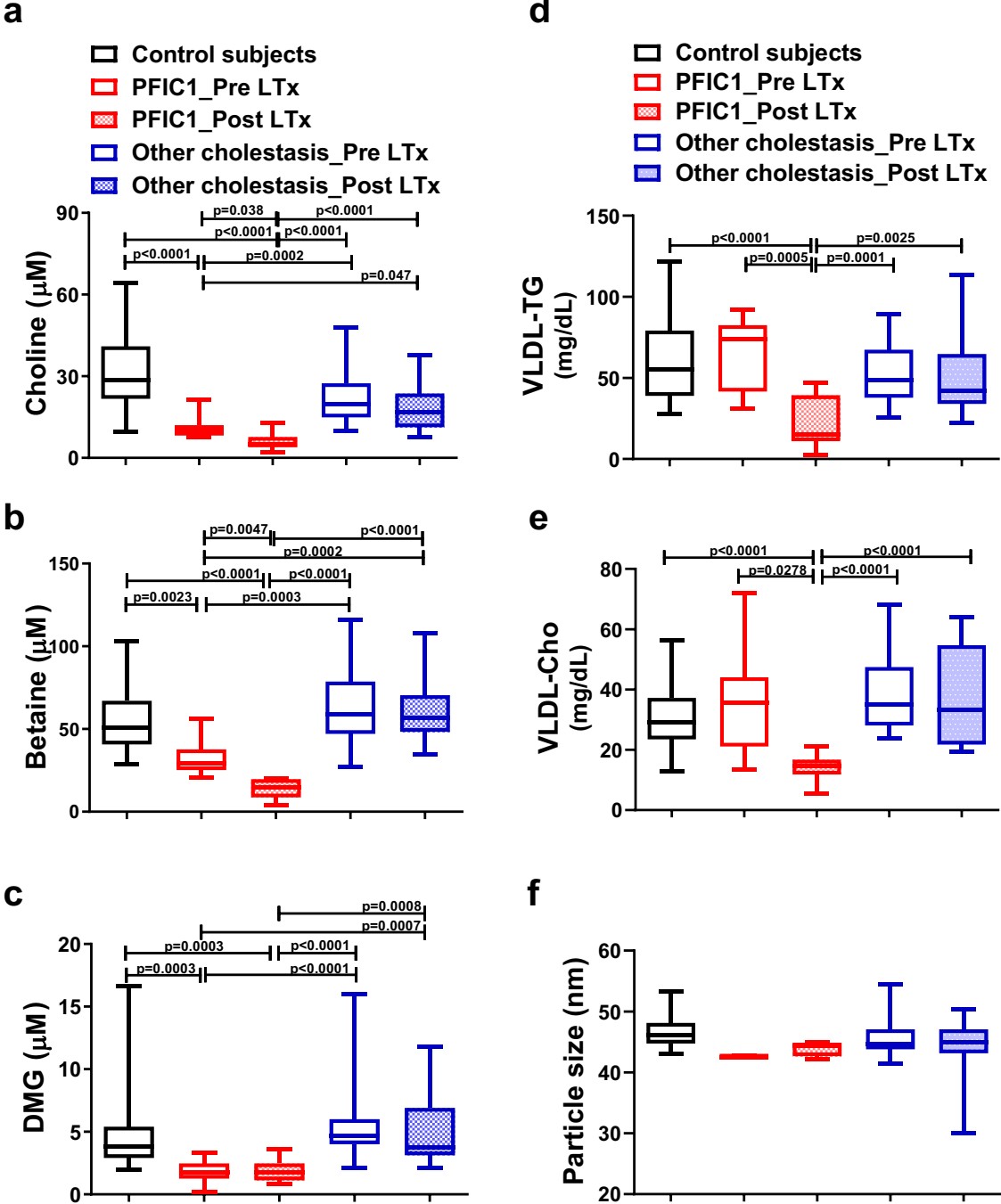

**Fig. 7 | PFIC1 patients present with decreased plasma concentrations of choline and its metabolites.** Fasting plasma was collected from PFIC1 patients with ($n = 12$) or without LTx ($n = 10$), other cholestatic patients with ($n = 20$) or without LTx ($n = 27$), and age-matched control subjects ($n = 31$). Demographic information of the patients is summarized in Table 2 and Supplementary Tables 2 and 3. **a**–**c** Plasma concentration of choline (**a**), betaine (**b**), and DMG (**c**). The collected specimens were analyzed to measure choline and its metabolites by LC/MS/MS. **d**–**f** Characterization of plasma VLDL. The collected specimens were analyzed to

evaluate the content of TG (**d**) and cholesterol (**e**) in VLDL and particle size (**f**) of VLDL using gel filtration high-performance liquid chromatography. Box-and-whisker plots are provided; the central line denotes the median value, the edges represent the upper and lower quartiles, and the whiskers indicate the minimum and maximum values. *P* values were calculated by Welch's one-way ANOVA with a post hoc Dunnett's T3 test for multiple comparisons and are indicated in the figures if less than 0.05. DMG dimethylglycine.

and its related metabolites is not the direct cause of these phenotypes. The primary function of the SI is to absorb nutrients. Growth retardation and high infant mortality rate have been shown in mice with impaired absorption of various nutrients, including glucose[52], amino acids[53], lipids[43], and vitamins[54]. However, in the metabolomic analysis, these possible candidates were not significantly altered in the plasma and IEC of Atp8b1$^{IEC-KO}$ mice. Accordingly, growth retardation and the

high infant mortality rate associated with Atp8b1 deficiency in IEC are still important issues that need to be addressed in future work.

In summary, Atp8b1 mediates LPC absorption in IEC to maintain hepatic choline and its metabolites at the normal level. Atp8b1$^{IEC-KO}$ mice developed steatohepatitis that CSD could completely resolve. The translational potential of these findings was suggested in the analysis of biological samples from PFIC1 patients. No therapy has been established

for PFIC1. LTx solves cholestasis and liver failure in PFIC1[18], but often results in insufficient clinical outcomes in PFIC1 because of graft steatohepatitis that can lead to graft liver failure[23,24]. Therefore, clinical trials should be conducted to test the efficacy and safety of choline supplementation therapy for steatohepatitis in PFIC1. If positive outcomes are confirmed, choline supplementation therapy could be a therapeutic option for PFIC1 and prevent the development and progression of steatohepatitis. Future investigation to understand the mechanism underlying the expression and function of ATP8B1 in IEC should reveal the group of diseases associated with steatohepatitis due to ATP8B1 dysfunction other than PFIC1 and expand the potential of choline supplementation therapy. In this study, we gave CSD before the onset of steatohepatitis in Atp8b1[IEC-KO] mice, so it is still being determined whether it would be effective after disease progression. Before clinical trials, the optimal timing to start choline administration must be examined.

## Methods

Additional details are provided in Supplementary Methods. The reagents used in this study were of analytical grade and are listed in Supplementary Table 4.

### Ethical approval

All mouse experiments were approved by and performed in accordance with the guidelines of the animal experiment committee of the University of Tokyo and the University of Tsukuba (permission number: P29-24). The study on human subjects was approved by the institutional review boards at the University of Tokyo, Juntendo University Graduate School of Medicine, Miyagi Children's Hospital, Tsuyama-Chuo Hospital, Yamaguchi University Graduate School of Medicine, Saiseikai Yokohama City Eastern Hospital, Kyoto University Hospital, and National Center for Child Health and Development (permission number: 24-5) and performed in accordance with the 1964 Declaration of Helsinki and its later amendments or comparable ethical standards (as revised in Edinburgh 2000). IRB approval covered the use of the control samples as well. Written informed consent was obtained from all subjects or their parents (when the subjects were under 18) before enrollment in the study.

### Animals and diets

To generate *Atp8b1[flox/flox]* mice, ICR and C57BL/6J mice were purchased from Charles River Laboratories International, Inc. (Yokohama, Japan). Mice were kept in plastic cages under pathogen-free conditions in a room maintained at 23.5 °C ± 2.5 °C and 52.5% ± 12.5% relative humidity under a 12-h light:12-h dark cycle. Mice had free access to a standard chow diet (D10012G; Research Diets Inc., New Brunswick, NJ) and filtered water. For choline supplementation in pups, mothers were fed a 1.0% CSD (D20011404; Research Diets Inc.) until the weaning of their pups.

### Mice generation

Two mouse genomic sequences (5′-GCAAGTGACTCAGACGTATG-3′ and 5′- CAGCTTGTAAGATCGCCTTG-3′) in introns 2 and 3 of *Atp8b1* were selected as guide RNA targets. We inserted each sequence into a *pX330-mC* plasmid, which carried both guide RNA and Cas9-mCdt1 expression units[55]. The flox donor plasmid DNA, *pflox-Atp8b1*, carried the genomic region from 2104 bp upstream to 1769 bp downstream of exon 3 of *Atp8b1*. Two loxp sequences were inserted into 908 bp upstream and 832 bp downstream of exon 3 of *Atp8b1* in this donor vector. The above DNA vectors were isolated with FastGene Plasmid Mini Kit (Nippon Genetics, Tokyo, Japan) and filtered by MILLEX-GV® 0.22 μm Filter unit (Merk Millipore, Darmstadt, Germany) for microinjection.

Pregnant mare serum gonadotropin (5 units) and human chorionic gonadotropin (5 units) were injected intraperitoneally into female C57BL/6J mice with a 48-h interval, and mated with male C57BL/6J mice. We collected zygotes from oviducts in mated females, and a mixture of two *pX330-mC* (circular, 5 ng/μL, each) and *pflox-Atp8b1* (circular, 10 ng/μL) was microinjected into zygotes. Subsequently, surviving zygotes were transferred into oviducts in pseudopregnant ICR females, and newborns were obtained.

To confirm the designed flox mutation, the genomic DNA from the tail of the newborns was purified with PI-200 (Kurabo Industries Ltd, Osaka, Japan) in accordance with the manufacturer's protocol. Genomic PCR was performed with KOD-Fx (Toyobo, Osaka, Japan). The primers (*Atp8b1* dflox long LeF: 5′-CGGGCTGCAGGAATTT-GATGGCTATTTACATTCCCACCGTCTA-3′ and *Atp8b1* dflox long RiR: 5′-ACGCATGTTTGAACGTCATGTCTAATTG-3′) were used for checking the correct flox and large deletion mutations. In addition, we checked the random integration of *pX330-mC* and *pflox-Atp8b1* by PCR with ampicillin resistance gene-detecting primers (Amp detection-F: 5′-TTGCCGGGAAGCTAGAGTAA-3′, and Amp detection-R: 5′-TTTGCCTTCCTGTTTTTGCT-3′) and confirmed that no founder carried the random integration allele.

Two mice harboring the designed floxed Atp8b1 allele were selected (line #12, male, and line #51, female) and bred with C57BL/6 mice. The heterozygous floxed (Atp8b1[flox/+]) mice in the obtained offspring were intercrossed to produce homozygous floxed (Atp8b1[flox/flox]) mice. Atp8b1[flox/flox] mice were crossed with villin-Cre transgenic mice (Stock No. 021504; The Jackson Laboratory, Bar Harbor, ME), and Atp8b1[flox/+];villin-Cre mice in the obtained offspring were mated with Atp8b1[flox/flox] mice to generate Atp8b1 [flox/flox];villin-Cre (Atp8b1[IEC-KO]) mice. The resulting Atp8b1[IEC-KO] mice and littermate Atp8b1[flox/flox] mice were used for animal experiments. See the supporting information for details of the genotyping protocol (Supplementary Fig. 1a, b).

Atp8b1[flox/flox] mice (line #12) were crossed with villin-Cre/ERT2 transgenic mice (Stock No: 020282; The Jackson Laboratory), and Atp8b1[flox/+]; villin-Cre/ERT2 mice in the obtained offspring were mated with Atp8b1[flox/flox] mice (line #12) to generate Tax inducible IEC-specific Atp8b1 knockout (Atp8b1[Tax-iIEC-KO]; Atp8b1 [flox/flox]; villin-Cre/ERT2) mice. The resulting Atp8b1[Tax-iIEC-KO] mice and littermate Atp8b1[flox/flox] mice were used for animal experiments. See the supporting information section for details of genotyping protocol (Supplementary Fig. 1a, b). Tax solution (10 mg/mL) was prepared by dissolving 20 mg Tax (Sigma-Aldrich, St. Louis, MO) in 0.2 mL EtOH and further diluted to 1.8 mL with corn oil (FUJIFILM Wako Pure Chemical, Osaka, Japan). Eight-week-old Atp8b1[Tax-iIEC-KO] mice and littermate Atp8b1[flox/flox] mice were treated daily with 1 mg Tax intraperitoneally and then analyzed.

### Tissue sampling from mice

Newborn and 4-week-old mice were anesthetized with isoflurane (2%, inhalation anesthesia apparatus), followed by a laparotomy. After tissue collection, they were euthanized by cervical dislocation. Blood from the inferior vena cava was collected into ethylenediaminetetraacetic acid (EDTA)-coated tubes. The plasma was separated by centrifugation at $1700 \times g$ for 15 min, snap-frozen in liquid nitrogen, and stored at −80 °C. Livers were harvested, snap-frozen in liquid nitrogen, and stored at −80 °C or were processed for histological microscopic analysis, as described in each section. SI was excised, rinsed with cold PBS containing 0.5 mM taurocholic acid, and processed for histological analysis as described in each section or for preparing IEC. Except for the brush border membrane fraction preparation, TTDR Kit (Becton Dickinson, Becton Drive Franklin Lakes, NJ) was used to isolate IEC from SI. In accordance with the manufacturer's instructions, SI was opened longitudinally, sliced into small fragments roughly 2 mm wide, incubated with 1×TTDR diluted in Dulbecco's modified Eagle's medium (DMEM; Thermo Fisher Scientific, San Jose, CA) for 30 min at 37 °C, mixed with 2 mM EDTA/PBS containing 1% BSA, filtered through a 70 μm cell strainer (Corning Inc., Corning, NY), and centrifuged at $250 \times g$ for 10 min at 4 °C. The cell pellet was incubated with 1×Pharm Lyse for 15 min at room temperature to lyse red blood

cells, mixed with 2 mM EDTA/PBS containing 1% BSA, and centrifuged at $250 \times g$ for 10 min at 4 °C. The prepared IEC were subjected to the flippase assay or snap frozen in liquid nitrogen and stored at −80 °C.

## Human specimens

A nationwide Japanese survey has been conducted to identify patients with PFIC since 2015. The clinical diagnosis of PFIC was based on the presence of unremitting hepatocellular cholestasis with intractable pruritus, jaundice with conjugated hyperbilirubinemia, and elevated serum bile-acid concentrations. Full physical examination, serological, viral, and metabolic markers measurements, imaging, and urine screening were performed to rule out other causes of cholestasis, including hepatitis B and C virus infections, inborn errors in bile-acid synthesis, and ductal origin. These patients were subjected to genetic testing to analyze all exons and flanking intron–exon boundaries of the genes responsible for neonatal/infantile intrahepatic cholestasis, including *ATP8B1*, by Sanger sequencing[31,56], and/or targeted next-generation sequencing[57]. Patients who carried disease-causing mutations in both alleles of *ATP8B1* and/or showed ATP8B1 deficiency in phenotypic analysis using peripheral blood monocyte-derived macrophages were diagnosed with PFIC1[31]. Twenty-two PFIC1 patients and 47 patients with a pathogenic variant in genes other than ATP8B1 were identified and enrolled in this study.

Peripheral blood samples from these patients and 30 age-matched control subjects, who were hospitalized for reasons other than liver diseases, were collected in EDTA-2K-coated blood sampling tubes (Becton Dickinson) for plasma isolation. The prepared specimens were stored at −80 °C until the measurement of choline metabolites and evaluation of the VLDL profile. Demographic information on the participants is summarized in Table 2 and Supplementary Tables 2 and 3.

## Quantitative PCR (qPCR)

RNA was isolated from mouse IEC and liver using ISOGEN2 (Nippon Gene, Tokyo, Japan) and subjected to a reverse-transcription reaction by ReverTra Ace® qPCR RT Master Mix with gDNA Remover (Toyobo). The prepared cDNA was analyzed to evaluate mRNA expression levels of target genes by real-time qPCR using a CFX Connect Real-Time System (Bio-Rad Laboratories, Hercules, CA), the appropriate software (CFX Maestro 1.1; Bio-Rad), and Thunderbird SYBR qPCR Mix (Toyobo). The primer sequences used in this study are summarized in Supplementary Table 5. Gene expression for each reaction was normalized by the expression of 18S rRNA.

## Biochemistry

Plasma levels of ALT, AST, T-bil, D-bil, and total bile acids were determined by automated measurement with DRI-CHEM (FUJIFILM, Tokyo, Japan) or by outsourcing to Oriental Yeast (Tokyo, Japan). The plasma VLDL profile was obtained by outsourcing to Skylight Biotech (Akita, Japan). Liver lipids were extracted by Bligh and Dyer method and subjected to enzymatic determination of TG and PC (FUJIFILM Wako Pure Chemical).

## Histological analysis and immunohistochemistry

Mouse liver was cut into small pieces, fixed with 10% neutral buffered formalin, and embedded in paraffin for H&E and IHC staining and in OCT compound (Sakura Finetek, Tokyo, Japan) for Oil red O staining. Mouse SI was rinsed with PBS and then 10% neutral buffered formalin, cut into two segments (corresponding to proximal and distal regions), opened longitudinally, rolled with a toothpick, kept in place with a pathological pin, then fixed with 10% neutral buffered formalin, and then embedded in paraffin.

Three-micrometer paraffin sections were prepared, deparaffinized, rehydrated, stained with H&E, and mounted in Entellan New (Merck, Branchburg, NJ). For IHC, the rehydrated paraffin sections were subjected to antigen retrieval with 1×Tris-HCl (pH 10) (Sigma-Aldrich, St. Louis, MO) in Decloaking Chamber NxGen (Biocare Medical, Concord, CA) for 20 min. Nonspecific binding was blocked with 3% BSA/PBS at room temperature for 1 h. Sections were stained with primary antibodies overnight at 4 °C followed by Alexa Fluor secondary antibody at room temperature for 1 h. The primary and secondary antibodies used are listed in Supplementary Table 6. After mounting with ProLong Diamond Anti-fade Mountant (Thermo Fisher Scientific), microscopic images were obtained by a Zeiss LSM 880 with Airyscan (Carl Zeiss, Jena, Germany) or Zeiss Axio Scan Z1 (Carl Zeiss) and processed on Zen 3.0 software (Carl Zeiss). To quantify the immunostained areas, digital images were analyzed by ImageJ software (ver. 1.53c; National Institutes of Health, Bethesda, MD).

Seven-micrometer cryosections were prepared, washed with PBS to remove OCT, incubated in 60% isopropanol for 1 min, stained with 0.5% Oil red O for 10 min, and washed with 60% isopropanol for 1 min and PBS for 1 min, and then mounted in 90% glycerol.

## Metabolomic and lipidomic analysis

Metabolites were extracted from plasma, liver, and IEC. Metabolite profiles were acquired by hydrophilic interaction liquid chromatography/tandem mass spectrometry (HILIC/MS/MS), gas chromatography/tandem mass spectrometry (GC/MS/MS) analyses, and non-targeted lipidomic analysis. Welch's *t* test was employed to detect group differences for each lipid and metabolite. The Benjamini–Hochberg method was applied for multiple testing corrections. The result of the lipidomic analysis was evaluated using Kolmogorov–Smirnov (KS) running sum statistic to determine the enrichment of lipid subtypes. Pathway-level analysis for choline metabolites was executed using the generally applicable gene set enrichment (GAGE) method[58]. The details are described in Supplementary Materials.

## Measurement of choline and methionine metabolites

The plasma concentrations of choline, betaine, DMG, methionine, SAM, homocysteine, and *S*-adenosyl-homocysteine were determined with an LC/MS/MS assay as described in Supplementary Materials.

## Cell culture

HEK293T (CRL-11268) cells and CHO-K1 (CCL-61) cells were purchased from the American Type Culture Collection (Manassas, VA). HEK293T cells were maintained in DMEM (Thermo Fisher Scientific) supplemented with 10% fetal bovine serum (FBS; Thermo Fisher Scientific) and 1% penicillin–streptomycin (FUJIFILM Wako Pure Chemical) and CHO-K1 cells were maintained in DMEM/F-12 (Thermo Fisher Scientific) supplemented with 10% FBS and 1% penicillin–streptomycin. Both cell lines were cultured at 37 °C in 5% $CO_2$ at 95% humidity. Both cell lines were transfected with the plasmids (pShuttle-EV or pShuttle-ATP8B1-FLAG) using PEI MAX (Polysciences, Warrington, PA) and used for both the flippase assay and the cytotoxicity assay as described in Supplementary materials. No commonly misidentified cell lines were used in this study. All cell lines were negative for mycoplasma contamination.

## Statistical analysis

Graphs include mean ± standard error of the mean (SEM), unless otherwise indicated. Differences between two and multiple variables were assessed at the 95% confidence level using Welch's *t* test and one-way analysis of variance (ANOVA) with post hoc Dunnett's T3 tests, respectively. The Kaplan–Meier method determined each genotype's survival curve. The data were analyzed using GraphPad Prism 9.3.1 (GraphPad Software, La Jolla, CA).

For metabolomic and lipidomic analyses, Welch's *t* test was employed to detect group differences for each lipid and metabolite. The Benjamini–Hochberg method was applied for multiple testing corrections. Data were analyzed with scipy (ver 1.11) and statsmodels (ver 0.14.0) modules of Python 3.

**Reporting summary**

Further information on research design is available in the Nature Portfolio Reporting Summary linked to this article.

## Data availability

Source data of each figure used in this study are available in the Supplementary Tables. The raw mass spectrometry data of the metabolomic and lipidomic analysis have been deposited onto Metabolights under accession number MTBLS8693. Source data are provided with this paper.

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

## Acknowledgements

We thank the patients and their families for their contributions to this study. We would also like to thank Minako Sakai and Yui Kubota (The University of Tokyo) for technical assistance in animal experiments, Takeo Moriya, Megumi Hirayama, and Chisato Kurosawa (Axcelead Drug Discovery Partners, Inc.) for supporting metabolomic and lipidomic analysis, and Edanz (https://jp.edanz.com/) for English language editing. This work was supported by Japan Agency for Medical Research and Development (AMED) grant JP21ek0109408 and 23ek0109580, Mochida Memorial Foundation for Medical and Pharmaceutical Research, and Takeda Science Foundation to H.H. The funding source did not participate in the study design and execution.

## Author contributions

R.T. and Y.S. performed experiments and analyzed and interpreted the experimental results. T.M. analyzed and interpreted the experimental results. S.M. and S.T. generated Atp8b1flox/flox mice. S.N., M.S., D.A., S.K., Y.A., A.I., T.O., S. Shimizu, A.F., S. Sakamoto, and M.K. collected clinical information and biological samples from pediatric patients and contributed to the interpretation of the experiments using them. Y.Z. contributed to the interpretation of the histological analysis. T.A. designed and performed metabolomic and lipidomic analysis and analyzed and interpreted the results. H.H. conceived and supervised the study and contributed to the design and interpretation of all experiments. R.T., Y.S., T.M., S.M. and H.H. drafted the manuscript. H.K., S.T., Y.Z., T.A. and H.H. revised the manuscript. All authors approved the final version of the manuscript.

## Competing interests

H.H., Y.S. and R.T. are coinventors on a patent (PCT/JP2022/44650) on therapy with choline metabolites for diseases caused by ATP8B1 dysfunction. The other authors declare no competing interests.
