## [Peer Review File · Nature Communications]

Intestinal Atp8b1 dysfunction causes hepatic choline deficiency and steatohepatitisREVIEWER COMMENTS

Reviewer #1 (Remarks to the Author):

This article analyses the physiological function and mechanism of ATP8B1, one of the causative genes of progressive familial intrahepatic cholestasis (PFIC), in mice (Villin-cre x *Atp8b1* flox mice). These mice showed early death, abnormal morphology of the small intestine and fatty liver. There was also an accumulation of lysophosphatidylcholine (LPC) in the intestinal epithelial cells and a decrease in choline and its metabolites in the liver and other organs. The authors have shown that reduced uptake of LysoPC from the luminal side into the cell membrane of *Atp8b1*-deficient intestinal epithelial cells, and thus reduced choline and phosphatidylcholine (PC) in the liver, results in reduced extrahepatic fat transport (i.e. fatty liver). It also confirms that fatty liver in these mice can be ameliorated by increasing the amount of choline in the diet. On the other hand, early death and morphological abnormalities of the small intestine are not improved by choline supplementation. As human PFIC patients also show a decrease in serum choline, we propose that the fatty liver observed in PFIC patients after liver transplantation may be improved by choline supplementation, thus increasing the success rate of transplantation. The study analyzed the function of the *Atp8b1* gene in the intestinal epithelium, the cell with the highest expression of the *Atp8b1* gene, which is important because it provides clues to the mechanism of PFIC development and treatment. The data are generally clear and strong enough to support the main argument. However, the fact that *Atp8b1* in intestinal epithelial cells is not associated with bile duct congestion, a major manifestation of PFIC, and that the mechanisms of early death and small intestinal morphological abnormalities in Villin-Cre *Atp8b1* flox mice remain unclear somewhat reduces the importance of this paper.

Questions and comments.

1. Regarding the data on choline and its metabolites in the liver in Fig. 4d and the extended data in Fig. 9a, the degree of reduction by IEC-KO seems to be quite different. Which is more accurate and why are both reported?
2. in Fig. 4d, it would be desirable to have data on the liver concentrations of PC and LPC, which are the key substances in this paper; data on PC are available in Fig. 6l, but were they

not measured during the experiment in Fig. 4?

3. in the figure of Extended Data Fig. 11, the amount of choline in the blood of PFIC1 is normal, but in the text Fig. 7a it is decreased; is the explanation of Ext. Fig. 11 incorrect?

4. in the mouse model, choline supplementation does not improve survival. Are symptoms due to intestinal abnormalities not a major problem in patients with successful liver transplantation?

Reviewer #2 (Remarks to the Author):

Dear authors,

The manuscript's title "Intestinal Atp8b1 dysfunction causes hepatic choline deficiency and steatohepatitis" demonstrated that Atp8b1 mediated LysoPC absorption in the intestinal epithelial cells (IEC) is dominant in maintaining normal hepatic choline and IEC-specific Atp8b1-knockout (Atp8b1IEC-KO) mice exhibit intestinal and hepatic phenotypes comparable to those of their littermate controls when newborns, but progress to steatohepatitis by 4 weeks. Atp8b1 deficiency in IEC causes LPC malabsorption and thereby hepatic choline deficiency. Complete recovery from steatohepatitis in 4-week-old Atp8b1IEC-KO mice is achieved by feeding choline-supplemented diets to lactating mice. Authors conclude that Choline supplementation therapy may be therapeutic for steatohepatitis caused by ATP8B1 dysfunction. Authors used the expected standard methodology.

I would like to recommend that this manuscript would be accepted to publish in the Nature Communication journal.

Reviewer #3 (Remarks to the Author):

This is a review report on the manuscript entitled "Intestinal Atp8b1 dysfunction causes hepatic choline deficiency and steatohepatitis", which has been submitted for potential publication in the Nature Communications journal.

In this study, mice lacking ATP8B1 phospholipid flippase in the intestinal epithelial cells (IEC-specific Atp8b1 knockout (Atp8b1IEC-KO) mice) were generated to evaluate the physiological function of Atp8b1 in IEC. In humans, a mutation in the gene encoding ATP8B1

cause progressive familial intrahepatic cholestasis type 1 (PFIC1), an extremely rare inherited autosomal recessive liver disease. In *Atp8b1*IEC-KO mice, a loss of *Atp8b1* mRNA in IEC and normal expression in the liver was confirmed by qPCR. The main findings regarding the function of *Atp8b1* and the consequences of its loss in IEC of animals are as follows:

- *Atp8b1*IEC-KO mice die within 12 weeks after birth and develop growth retardation and elongation of the small intestine by 4 weeks of age.
- By the age of 4 weeks, KO mice develop a liver injury, which was reflected by the higher levels of AST, ALT, and bilirubin in blood as well as several other parameters measured with IHC staining of the liver sections.
- Metabolomics and lipidomics analyses of IEC, plasma, and liver from *Atp8b1*IEC-KO mice and their *Atp8b1*flox/flox littermates at 4 weeks of age showed that, in KO mice, LPC species are elevated in IEC, while choline metabolites decreased in the liver. TG level was found elevated in the liver of KO animals. These observations are typical of steatohepatitis.
- *Atp8b1*IEC-KO and *Atp8b1*flox/flox mice had similar faecal content of PC, GPC, and choline which indicate a normal ability to digest PC in the intestinal lumen and absorb choline and GPC in the IEC of KO mice. It was also confirmed based on the experiment with [³H]-choline administered orally to studied animals.
- The flipping activity of ATP8B1 was confirmed with a cell-based flippase assay performed using nitrobenzoxadiazole (NBD)-labelled phospholipids and flow cytometry. This part was performed using CHO-K1 and HEK293T cells with exogenous ATP8B1 expression. Flippase assay with IEC indicated that ATP8B1 mediates the incorporation of NBD-LPC and NBD-PC into the inner leaflet of the plasma membrane of IEC.
- *Atp8b1* defect in IEC affects LPC absorption leading to hepatic choline deficiency and consequent development of steatohepatitis.

Based on these observations, the authors hypothesized that choline supplementation may prevent steatohepatitis in *Atp8b1*IEC-KO mice. Breastfeeding mice were supplemented with choline during lactation. Supplementation did not improve the survival rate, growth retardation, or greater weight of the SI in *Atp8b1*IEC-KO mice. Still, it relieved the greater length of the SI, led to a heavier liver weight, restored hepatic PC deficiency, and completely suppressed the development of steatohepatitis in these animals.

Finally, plasma samples of patients with PFIC1 (pre- and post-liver transplantation), with other cholestatic diseases and from age-matched controls were analyzed. In PFIC1 patients, deficiency in choline and its metabolites was observed. PFIC1 patients after liver transplantation had abnormal hepatic VLDL secretion, which is consistent with the fact that steatohepatitis often develops in the graft liver of PFIC1 patients.

This valuable study explains the mechanisms by which *Atp8b1* loss in IEC may lead to hepatic choline deficiency and consequent steatohepatitis. The study is complex, complete and well-designed. The applied methodology is adequate to the questions asked. Have you tested the normality of data distribution before choosing parametric methods for the statistical analysis? Regarding metabolomics/lipidomics results, have you corrected obtained p-values for multiple comparisons? In the case of human data, the number of samples may be sufficient to achieve normal data distribution. Still, in the case of animal data, non-parametric tests should be used (maybe with some exceptions with high n). Consequently, data should be presented as median and IQR when applying non-parametric tests. I can't entirely agree with the author's statement that the results obtained from human samples confirm the translational potential of the study. These results confirm that choline metabolism, as evaluated in plasma, is similar in PFIC1 patients and *Atp8b1*IEC-KO mice. The results clearly show that clinical trials with choline supplementation should be performed on PFIC1 patients, especially those after liver transplantation. And I believe a positive outcome of such a clinical trial will have translational potential.

My other minor comments are:

Abstract - Please modify the second sentence to "Dietary phosphatidylcholine (PC) is digested into lysoPC (LPC), glycerophosphocholine (GPC), and choline in the intestinal lumen and is the primary source of choline in the body".

Page 6, lines 20-22 – Please clearly indicate in this sentence that the elevation of LPC was observed in IEC.

Reviewer #1

We are grateful to the reviewer for raising important issues and providing helpful comments and suggestions. Our replies to the reviewer's queries and the revisions to the text are as follows.

Questions and comments:

1. Regarding the data on choline and its metabolites in the liver in Fig. 4d and the extended data in Fig. 9a, the degree of reduction by IEC-KO seems to be quite different. Which is more accurate and why are both reported?

Answer: The data in Supplementary Fig. 9a is more accurate than that in Fig. 4d. The data in Supplemental Fig.9a was obtained by absolute calibration curve, whereas Fig. 4a was data of metabolomic analysis obtained by a relative comparison between control and IEC-KO mice. Fig.4 illustrates how exploratory studies using metabolomic and lipidomic analyses led to the hypothesis that IEC-KO mice develop steatohepatitis due to hepatic choline deficiency caused by LPC malabsorption. Supplementary Fig.9 verifies the hepatic choline deficiency in IEC-KO mice inferred from the results of Fig.4 with a more accurate analytical method. We have corrected the sentence as follows:

"Regardless of the length and degree of unsaturation of fatty acid, almost all LPC species detected by the lipidomic analysis were elevated in IEC of *Atp8b1*^{IEC-KO} mice (line #12) (Fig. 4b). This finding was confirmed by absolute quantification of LPC species in the brush border membrane fractions prepared from IEC (Supplementary Fig. 5a, b). Metabolomic analysis showed a trend toward decreased choline metabolites in the plasma and liver of *Atp8b1*^{IEC-KO} mice (line #12) compared to the littermate *Atp8b1*^{fl^{ox}/fl^{ox}} mice (Fig. 4 c–e). The pathway-level analysis of metabolomic data focused on choline metabolism detected a statistically significant deficiency of choline metabolites in the plasma and liver of *Atp8b1*^{IEC-KO} mice (line #12) (Table 1). Absolute quantification using LC/MS/MS optimized for detecting choline metabolites confirmed that *Atp8b1*^{IEC-KO} mice (line #12) had lower hepatic levels of choline and its metabolites than the littermate *Atp8b1*^{fl^{ox}/fl^{ox}} mice (Supplementary Fig. 9a)." (page 6, line 19–page 7, line 2)

"For identified metabolites, MRM peak areas were calculated, and relative comparisons were made between samples." (Supplemental material; page 3, line 18–page 4, line 1; page 4, lines 11–12; and page 5, lines 7–8)

2. in Fig. 4d, it would be desirable to have data on the liver concentrations of PC and LPC, which are the key substances in this paper; data on PC are available in Fig. 6l, but were they not measured during the experiment in Fig. 4?

Answer: Fig. 4 was created to illustrate how exploratory studies using metabolomic and lipidomic analyses led to the hypothesis that IEC-KO mice develop steatohepatitis due to hepatic choline deficiency caused by LPC malabsorption. We extracted compounds whose concentrations in IEC, plasma, or liver differed between control and IEC-KO mice. Of the extracted compounds, those relevant to our hypothesis are shown in Fig. 4. In the lipidomic analysis employed in this study, PC and LPC are the molecules to be measured. No statistical difference

was detected between control and IEC-KO mice in liver concentrations of either compound group. This is most likely because the lipidomic analysis applies relative comparisons, a less accurate method than the absolute quantification employed in Fig. 6l. Deficiency of choline metabolites in the liver of IEC-KO mice has been demonstrated by the absolute quantitative methods, as shown in Fig. 6l and Supplementary Fig.9a.

3. in the figure of Extended Data Fig. 11, the amount of choline in the blood of PFIC1 is normal, but in the text Fig. 7a it is decreased; is the explanation of Ext. Fig. 11 incorrect?

Answer: We apologize for the incorrect description of blood choline levels in Supplementary Fig. 11. We have corrected Supplementary Fig. 11.

4. in the mouse model, choline supplementation does not improve survival. Are symptoms due to intestinal abnormalities not a major problem in patients with successful liver transplantation?

Answer: After liver transplantation, PFIC1 patients present with retardation of stature and weight as well as steatosis and fibrosis in the graft liver, both of which are major clinical problems. The intestinal abnormalities could contribute to their physical growth retardation. The phenotypes of IEC-KO mice support this. The choline supplementation diet failed to alleviate the growth retardation and high infant mortality rate in IEC-KO mice. Thus, these are still clinically essential issues that should be addressed in future work. We have corrected the sentence as follows:

“However, it often results in insufficient clinical outcomes in PFIC1 because of exacerbation of steatosis and fibrosis in the graft liver and retardation of stature and weight^{23,24}.” (page 3, lines 25–26)

We sincerely hope that you will find these revisions and corrections satisfactory and that our paper will now be acceptable for publication in *Nature Communications*.

Reviewer #2

We are grateful to the reviewer for reviewing our manuscript. Our replies to the reviewer's comments are as follows.

Comments:

Authors used the expected standard methodology. I would like to recommend that this manuscript would be accepted to publish in the Nature Communication journal.

Answer: We are grateful to you for reviewing our manuscript.

We sincerely hope our paper will now be acceptable for publication in *Nature Communications*.

Reviewer #3

We are grateful to the reviewer for raising important issues and providing helpful comments and suggestions. Our replies to the reviewer's queries and the revisions to the text are as follows.

Comments:

1. Have you tested the normality of data distribution before choosing parametric methods for the statistical analysis? In the case of human data, the number of samples may be sufficient to achieve normal data distribution. Still, in the case of animal data, non-parametric tests should be used (maybe with some exceptions with high n). Consequently, data should be presented as median and IQR when applying non-parametric tests.

Answer: We apologize for the inadequate and incorrect description of statistical methods in the initially submitted manuscript. The normality of data distribution was not evaluated. This is because the results obtained are unreliable due to the limited sample size in this study. Standard nonparametric tests, such as Mann-Whitney U test, assume equal variances. However, when the sample size is limited, the equal-variance assessment is not reliable. Therefore, nonparametric tests were not used. This study used Welch test because, in recent years, Welch test has been recommended in cases where these risks exist (Stat Papers (2011) 52:219–231, DOI: 10.1007/s00362-009-0224-x). We have corrected the sentence as follows:

"Differences between two and multiple variables were assessed at the 95% confidence level using Welch's t-test and one-way analysis of variance (ANOVA) with post hoc Dunnett's T3 tests, respectively." (page 20, lines 16–18)

"*P < 0.05, ****P < 0.0001 by two-tailed, unpaired Welch's t-test." (page 30, line 10)

"*P < 0.05, ****P < 0.0001 by two-tailed, unpaired Welch's t-test." (page 31, line 12)

"*P < 0.05, **P < 0.01, ***P < 0.001, ****P < 0.0001 by two-tailed, unpaired Welch's t-test." (page 32, line 10)

"*P < 0.05 by two-tailed, unpaired Welch's t-test with the Benjamini–Hochberg correction (d). **P < 0.01 by two-tailed, unpaired Welch's t-test (g)." (page 34, lines 8–9)

"****P < 0.001 by two-tailed, unpaired Welch's t-test." (page 35, line 16)

"*P < 0.05, **P < 0.01, ***P < 0.001, ****P < 0.0001 by Welch's one-way ANOVA with a post hoc Dunnett's T3 test for multiple comparisons." (page 37, lines 16–17)

"Box-and-whisker plots are provided; the central line denotes the median value, the edges represent the upper and lower quartiles, and the whiskers indicate the minimum and maximum values. *P < 0.05, **P < 0.01, ***P < 0.001, ****P < 0.0001 by Welch's one-way ANOVA with a post hoc Dunnett's T3 test for multiple comparisons." (page 38, lines 8–11)

"***P < 0.01, ***P < 0.001, ****P < 0.0001 by two-tailed, unpaired Welch's t-test." (Supplemental material; page 14, lines 8–9)

"****P < 0.0001 by two-tailed, unpaired Welch's t-test." (Supplemental material; page 15, lines 10–11)

“*P < 0.05, ****P < 0.0001 by two-tailed, unpaired Welch’s t-test.” (Supplemental material; page 16, line 9)

“P < 0.05 by two-tailed, unpaired Welch’s t-test.” (Supplemental material; page 17, line 10)

“**P < 0.01, ****P < 0.0001 by two-tailed, unpaired Welch’s t-test.” (Supplemental material; page 18, lines 9–10)

“*P < 0.05, **P < 0.01, ***P < 0.001 by two-tailed, unpaired Welch’s t-test.” (Supplemental material; page 19, lines 9–10)

“**P < 0.01, ***P < 0.001, ****P < 0.0001 by two-tailed, unpaired Welch’s t-test.” (Supplemental material; page 20, lines 8–9)

“*P < 0.05, ***P < 0.001, ****P < 0.0001 by two-tailed, unpaired Welch’s t-test.” (Supplemental material; page 21, lines 9–10)

2. Regarding metabolomics/lipidomics results, have you corrected obtained p-values for multiple comparisons?

Answer: We apologize for the inadequate and incorrect description of statistical methods in the initially submitted manuscript. In the revised manuscript, Welch's t-test was employed to detect group differences for each lipid and metabolite. The Benjamini-Hochberg method was applied for multiple testing corrections. Metabolomic and lipidomic analyses were performed to obtain a hypothesis of the mechanism by which Atp8b1^{IEC-KO} mice develop steatohepatitis. Metabolomic analysis showed that choline metabolites in the plasma and liver of Atp8b1^{IEC-KO} mice tended to be reduced, compared to the littermate Atp8b1^{fl^{ox}/fl^{ox}} mice (Fig. 4c–e). The pathway-level analysis of metabolomic data focused on choline metabolism detected a statistically significant deficiency of choline metabolites in the plasma and liver of Atp8b1^{IEC-KO} mice (Table 1). The enrichment analysis of lipidome data revealed an increase in LPC in IEC of Atp8b1^{IEC-KO} mice (Fig. 4a). These results in exploratory studies were validated by the absolute quantification of LPC species in the IEC (Supplementary Fig. 5a, b) and choline metabolites in the liver (Supplementary Fig. 9a), which showed statistically significant differences between Atp8b1^{IEC-KO} mice and the littermate Atp8b1^{fl^{ox}/fl^{ox}} mice. Based on these findings, we concluded that Atp8b1^{IEC-KO} mice develop steatohepatitis due to hepatic choline deficiency caused by LPC malabsorption. We have added Table 1 and corrected the sentence as follows:

“Regardless of the length and degree of unsaturation of fatty acid, almost all LPC species detected by the lipidomic analysis were elevated in IEC of Atp8b1^{IEC-KO} mice (line #12) (Fig. 4b). This finding was confirmed by absolute quantification of LPC species in the brush border membrane fractions prepared from IEC (Supplementary Fig. 5a, b). Metabolomic analysis showed a trend toward decreased choline metabolites in the plasma and liver of Atp8b1^{IEC-KO} mice (line #12) compared to the littermate Atp8b1^{fl^{ox}/fl^{ox}} mice (Fig. 4c–e). The pathway-level analysis of metabolomic data focused on choline metabolism detected a statistically significant deficiency of choline metabolites in the plasma and liver of Atp8b1^{IEC-KO} mice (line #12) (Table 1). Absolute quantification using LC/MS/MS optimized for detecting choline metabolites confirmed that Atp8b1^{IEC-KO} mice (line #12) had lower hepatic levels of choline and its metabolites than the littermate Atp8b1^{fl^{ox}/fl^{ox}} mice (Supplementary Fig. 9a).” (page 6, line 19–page 7, line 2)

“Pathway-level analysis for choline metabolites was executed using the generally applicable gene set enrichment (GAGE) method ⁵⁸.” (page 19, lines 21–22)

“For metabolomic and lipidomic analyses, Welch's t-test was employed to detect group differences for each lipid and metabolite. The Benjamini-Hochberg method was applied for multiple testing corrections. Data was analyzed with scipy (ver 1.11) and statsmodels (ver 0.14.0) modules of python 3.” (page 20, lines 21–22)

“Pathway-level analysis for choline metabolites

Pathway-level analysis for choline metabolites was performed using the generally applicable gene set enrichment (GAGE) method ⁵⁸. Briefly, metabolites in the choline pathway and overall metabolites were subjected to Welch's t-test, comparing all sample combinations of Atp8b1^{IEC-KO} mice (line #12) and the littermate Atp8b1^{flox/flox} mice. Negative log-sum values of resultant p-values were adjusted based on control group dependencies, and the integrated p-value was computed on a Gamma distribution with K degrees of freedom and a scale of 1.0, where K represents the number of samples of Atp8b1^{IEC-KO} mice (line #12). Data was analyzed with scipy (ver 1.11) modules of Python 3.” (Supplemental material; page 6, line 16–page 7, line 4)

3. I can't entirely agree with the author's statement that the results obtained from human samples confirm the translational potential of the study. These results confirm that choline metabolism, as evaluated in plasma, is similar in PFIC1 patients and Atp8b1^{IEC-KO} mice. The results clearly show that clinical trials with choline supplementation should be performed on PFIC1 patients, especially those after liver transplantation. And I believe a positive outcome of such a clinical trial will have translational potential.

Answer: As pointed out by the reviewer, clinical trials evaluating the efficacy and safety of choline supplementation therapy for steatohepatitis in PFIC1 patients are essential to validate the translational potential of the findings in this study. This study only suggests its translational potential by analyzing biological samples from PFIC1 patients. We have corrected the sentence as follows:

“This study indicates that Atp8b1 regulates hepatic choline levels through intestinal LPC absorption, encouraging the evaluation of choline supplementation therapy for steatohepatitis caused by ATP8B1 dysfunction.” (page 2, lines 10–12)

“The translational potential of these findings was suggested in the analysis of biological samples from PFIC1 patients” (page 13, lines 22–23)

“Therefore, clinical trials should be conducted to test the efficacy and safety of choline supplementation therapy for steatohepatitis in PFIC1. If positive outcomes are confirmed, choline supplementation therapy could be a therapeutic option for PFIC1 and prevent the development and progression of steatohepatitis.” (page 13, line 26–page 14, line 2)

4. Abstract - Please modify the second sentence to "Dietary phosphatidylcholine (PC) is digested into lysoPC (LPC), glycerophosphocholine (GPC), and choline in the intestinal lumen and is the primary source of choline in the body".

Answer: Following the advice of the reviewer, we have corrected this sentence as follows:

"Dietary phosphatidylcholine (PC) is digested into lysoPC (LPC), glycerophosphocholine, and choline in the intestinal lumen and is the primary source of systemic choline." (page 2, lines 2–4)

5. Page 6, lines 20-22 – Please clearly indicate in this sentence that the elevation of LPC was observed in IEC.

Answer: Following the advice of the reviewer, we have corrected this sentence as follows:

"Regardless of the length and degree of unsaturation of fatty acid, almost all LPC species detected by the lipidomic analysis were elevated in IEC of *Atp8b1*^{IEC-KO} mice (line #12) (Fig. 4b)." (page 6, lines 19–22)

We sincerely hope that you will find these revisions and corrections satisfactory and that our paper will now be acceptable for publication in *Nature Communications*.

REVIEWERS' COMMENTS

Reviewer #1 (Remarks to the Author):

I am satisfied the responses and revisions.

Reviewer #3 (Remarks to the Author):

The authors have addressed all my comments and implemented adequate changes in the manuscript. I do not have further comments and in my opinion, the manuscript can be accepted for publication.

Reviewer #1

We are grateful to the reviewer for reviewing our manuscript. Our replies to the reviewer's comments are as follows.

Comments:

I am satisfied the responses and revisions.

Answer: We are grateful to you for reviewing our manuscript.

We sincerely hope our paper will now be acceptable for publication in *Nature Communications*.

Reviewer #3

We are grateful to the reviewer for reviewing our manuscript. Our replies to the reviewer's comments are as follows.

Comments:

The authors have addressed all my comments and implemented adequate changes in the manuscript. I do not have further comments and in my opinion, the manuscript can be accepted for publication.

Answer: We are grateful to you for reviewing our manuscript.

We sincerely hope our paper will now be acceptable for publication in *Nature Communications*.